

# Short Communication: Motivation for standardizing and normalizing inter-model comparison of computational landscape evolution models

Nicole M. Gasparini[1], Katherine R. Barnhart[2,3,4*], and Adam M. Forte[5,*]

[1]Tulane University, Earth and Environmental Sciences Department, New Orleans, LA, USA
[2]University of Colorado at Boulder, Cooperative Institute for Research in Environmental Sciences, Boulder, CO, USA
[3]University of Colorado at Boulder, Department of Geological Sciences, Boulder, CO, USA
[4]Now at U.S. Geological Survey, Geologic Hazards Science Center, Golden, CO, USA
[5]Department of Geology & Geophysics, Louisiana State University, Baton Rouge, LA, USA
[*]These authors contributed equally to this work.

**Correspondence:** Nicole Gasparini (ngaspari@tulane.edu)

**Abstract.** This manuscript is a call to the landscape evolution modeling community to develop benchmarks for model comparison. We illustrate the use of benchmarks in illuminating the strengths and weaknesses of different landscape evolution models (LEMs) that use the stream power process equation (SPPE) to evolve fluvial landscapes. Our examples compare three different modeling environments—CHILD, Landlab, and TTLEM—that apply three different numerical algorithms on three different grid types. We present different methods for comparing the morphology of steady-state and transient landscapes, as well as the time to steady state. We illustrate the impact of time step on model behavior. There are numerous scenarios and model variables that we do not explore, such as model sensitivity to grid resolution and boundary conditions, or processes beyond fluvial incision as modeled by the SPPE. Our examples are not meant to be exhaustive. Rather, they illustrate a subset of best practices and practices that should be avoided. We argue that all LEMs should be tested in systematic ways that illustrate the domain of applicability for each model. A community effort beyond this study is required to develop model scenarios and benchmarks across different types of LEMs.

## 1 Introduction

Thirty years ago there were only a handful of computational landscape evolution models (LEMs, Ahnert, 1976; Willgoose et al., 1991; Chase, 1992; Howard, 1994; Tucker and Slingerland, 1994). Now there are multiple review papers on the theory behind, differences among, and uses of different computational landscape evolution models (Coulthard, 2001; Martin and Church, 2004; Tucker and Hancock, 2010; van der Beek, 2013; Temme et al., 2013; Chen et al., 2014). Yet, even as the number of computational landscape evolution models and modeling environments proliferate, models become more sophisticated, and modeling becomes a standard tool for testing hypotheses in the geomorphology community, the practice of inter-model comparison is not generally practiced within the LEM community.





All inter-model comparisons seek to compare models in some systematic way. One common type of inter-model comparison is called a 'benchmark' and typically involves a known answer. This known answer may derive from the analytical solution to the model governing equations, or another approach, such as the method of manufactured solutions (Roache, 1998). The specification of benchmark problems are typically the result of community consensus. Model performance on benchmark problems indicates the model can reliably solve that type of problem. Benchmark problems typically have simple initial and boundary conditions such that multiple models can implement them.

Inter-model comparison is standard in the climate science community. Projects like the Coupled Model Intercomparison Project (CMIP) have been ongoing for decades (Meehl et al., 2000). (Here coupled model refers to global climate models (GCMs) that include energy fluxes among the ocean, atmosphere, cryosphere, and land.) The many iterations of CMIP have led to established scenarios for comparing model results and standardized formats of model output. Climate modeling has implications for policy and humanity. Thus the uncertainty and range in outcomes is highly scrutinized by the modeling community and the public.

However, inter-model comparison is not just a tool for the climate community. Other earth science modeling communities have embraced inter-model comparison. For example, Buiter et al. (2016) designed three numerical experiments for intercomparison of results among 11 different computational models of an accretionary wedge. If a researcher were to develop a new model, they could use the benchmark scenarios laid out in Buiter et al. (2016) to quantitatively test whether their model was providing reasonable solutions before applying it to new scenarios. Similarly, the community modeling dynamic earthquake rupture has produced a series of benchmark experiments and engaged in a process of inter-model verification of simulation results (e.g., Harris and Archuleta, 2004; Harris et al., 2009, 2011; Barall and Harris, 2015).

Some model comparison studies have been performed in the LEM community, but these have primarily focused on how different processes impact model outcome. For example, there have been multiple studies that used the same modeling framework but different fluvial process equations (e.g., Whipple and Tucker, 2002; Crosby et al., 2007; Gasparini et al., 2007; Attal et al., 2011) or hillslope process equations (e.g., Tucker and Bras, 1998; Roering, 2008; Barnhart et al., 2019) or both (Barnhart et al., 2020b) to quantify the topographic signature of different processes. Further, both Campforts et al. (2017) and Dietterich et al. (2017) compared topographic outcomes produced using different numerical methods to solve the same process equation within the same model. Process model sensitivity to parameters has been explored (e.g., Tucker and Whipple, 2002; Barnhart et al., 2020b; Shobe et al., 2017) as well as sensitivity to parameters and grid resolution (Kwang and Parker, 2017). All of these studies are intra-model comparisons, in that they used the same modeling environment for all of the model experiments. However, there have been no inter-model comparisons or benchmark experiments of LEMs of which we are aware. There are examples of two different models using the same algorithms and benchmarks for verification (de Almeida et al., 2012; Adams et al., 2017), but these were done in series, not as a coordinated effort. It is reasonable to expect that different models that implement the same process equation, with the same parameters and boundary and initial conditions would lead to the same outcomes, at least statistically. However, the LEM community has never laid out how to set-up benchmark studies and what should be quantitatively compared in the model outcomes.





Here, we present results that motivate the establishment of a community effort to generate benchmark experiments of LEMs.
We illustrate inter-model and intra-model comparisons of landscapes modeled using the stream power process equation (SPPE).
The SPPE is widely used in LEMs to describe fluvial incision as a power law function of drainage area and topographic slope
(e.g., Howard, 1994; Whipple and Tucker, 1999). We perform the same experiments in three different modeling environments.
In some cases, we compare the solution of the equation using different numerical algorithms in the same modeling environment.
We also compare the solution of the equation using different grid types. Finally, we compare solutions using the same numerical
algorithm and grid type but implemented in different modeling environments. When possible, we use the same initial condition
for the different models, and in all cases we use the same boundary conditions. We explore differences in model outcomes at
steady state (here defined as bedrock incision rate equal to rock uplift rate) and following a perturbation (increase in rock uplift
rate) after reaching steady state. The take-home message will not be surprising to anyone who has used a LEM; all numerical
algorithms have limitations. However, the differences and similarities among the algorithms lead to insights on how to properly
use these models and the need for established benchmarks for both current and future LEMs.

The goal of this paper is not to encourage users towards, or discourage users away from, the models we compare. It is also
not to argue that the specific set of comparisons here are the best choice of benchmarks. Rather, it is an initial illustration of one
vision of inter- and intra-LEM comparison intended to inspire community collaboration to establish benchmark problems for
LEMs and support future inter- and intra-model comparison studies. This project was a side project and labor of love from three
LEM community members. In contrast, other scientific model benchmarking studies were based on multiple funded meetings
and input from a much larger swath of the community. We hope such an effort can grow from the seed of this paper.

## 2 Terminology

We define some terms that are often used with different meanings among studies. However, the specific definition that we use
is important for understanding this study.

Boundary conditions: The rules that are applied around the perimeter of a modeling area so that numerical algorithms can be
solved on the edges of the model domain. For example, whether water and sediment can flow into and out of nodes on model
edges is a boundary condition.

Computer model: The application of a numerical algorithm or algorithms to solve process equations on a grid. In this case,
the computer model is a landscape evolution model. We sometimes shorten this to model for readability.

Geomorphic process: A physical process that is shaping the earth's surface, such as a landslide or abrasion of a bedrock
riverbed by saltating sediment.

Modeling environment: The entire source code from which an individual computer model is created. For example, TTLEM
is a modeling environment, but we implement different TTLEM computer models that use the different numerical algorithms
contained within TTLEM.



Node: A location on a model grid. Variables that are required to solve a process equation are assigned to every node (e.g. elevation) or the connection between nodes (e.g. in some computer models slope is defined between nodes, in others it is defined at a node).

Numerical algorithm: A numerical algorithm uses what are commonly known as numerical methods to solve a differential equation so that it can be applied in a computer model. For example, the implicit finite difference method can be used as a

numerical algorithm to calculate fluvial bedrock incision from the stream power process equation.

Process equation: An equation that describes a physical process and includes variables and parameters that can be measured, estimated, or calibrated. Examples of process equations in a LEM include sediment transport rate and bedrock incision rate equations. Process equations in geomorphology are often referred to as models because they are a simplification, or a model, of a natural process.

Simulation: A model run with a fully specified set of numerical algorithms, input parameters, and initial and boundary conditions.

We also include some scientific and computational terms with precise definitions, many of which were taken after Schlesinger et al. (1979); Roache (1998); Gerya (2010).

Accuracy: "the degree to which the result of a measurement, calculation, or specification conforms to the correct value or a

standard." (Oxford English Dictionary accessed via Google dictionary).

Benchmark: A numerical simulation with specified initial and boundary conditions, and a known solution. Benchmarking a computer model is one part of verification.

Domain of applicability: The set of conditions for which a computer model has been verified and validated for use.

Numerically stable: An algorithm is considered numerically stable when it does not amplify errors, such as approximation or

truncation errors. When an algorithm is unstable, the numerical solution diverges from the true solution, can oscillate between extreme positive and negative values, and can take on non-physical values.

Validation: Demonstration that a computer model agrees with reality within a stated level of accuracy. That is, is one using the correct model for the question at hand.

Verification: The demonstration that a computer model for a given system represents the stated conceptual model or set of

governing equations to a stated level of accuracy. That is, solving the stated model equations correctly.

## 3    Modeling environments and comparisons

The three modeling environments used in this paper were chosen because the authors have experience using them. The choice of these models says nothing about the value of these chosen modeling environments or the variety of other modeling environments that exist. Explicitly, our goal was not to develop a comprehensive inter- or intra-model comparison of these

environments or others, but to show one possible way such comparisons could be done through broader community effort and to explore why such comparisons are warranted and needed.





### 3.1 CHILD

The CHILD modelling environment was developed in the late 1990s (e.g., Tucker et al., 1999, 2001a, b). It operates on a triangular irregular network, forming a voronoi diagram, here referred to as a voronoi grid for consistency with the other grid types. The CHILD experiments in this study all use an explicit finite difference solution of the stream power process equation.

CHILD is a C++ code that is open source and available through GitHub (https://github.com/childmodel/child; accessed 08 September 2022). Although CHILD is no longer actively in development, it was widely used in the past 20 years and continues to be used at the time of writing. Because of its familiarity to the authors, its application on a voronoi grid, and its advanced stage of development, we use it in this study.

### 3.2 Landlab

Landlab is Python library for modeling surface processes on regular and irregular grids (Hobley et al., 2017; Barnhart et al., 2020a). In this study we implement computer models that use raster, hexagonal, and voronoi grids. (Landlab also supports radial grids.) Landlab is open source and available through GitHub (https://github.com/landlab/landlab; accessed October 14, 2022). This study used Landlab version 2.4.2.dev0. Landlab is in active development and is currently maintained through CSDMS (Tucker et al., 2022). The stream power process component used in all Landlab computer models in this study implements a version of the Fastscape implicit finite difference numerical algorithm (Braun and Willett, 2013) The Fastscape numerical algorithm was designed to be more stable than most finite difference methods (Braun and Willett, 2013), the details of which are briefly discussed later.

### 3.3 TTLEM

TTLEM is part of the TopoToolbox Matlab library. Many readers may be familiar with the DEM (digital elevation model) analysis tools that are contained within TopoToolbox (e.g., Schwanghart and Scherler, 2014). TTLEM is an LEM that is distributed with these tools and uses the TopoToolbox library for many of the core functions within the LEM, e.g., flow routing (Campforts et al., 2017) (Available at https://github.com/wschwanghart/topotoolbox; accessed December 7, 2022).

TTLEM contains three numerical algorithms for solving the stream power equation. In this study we use TTLEM computer models that implement the Fastscape implicit numerical algorithm, an explicit finite difference algorithm, and the total variation diminishing finite volume method (TVD_FVM). TVD_FVM is designed to be highly accurate and limit numerical diffusion (Campforts and Govers, 2015).

### 3.4 LEMs and comparison rational

Using the three modeling environments described above, we created seven different LEMs which we used for intra- and inter-model comparison. These different LEMs are described in Table 1.

These three modeling environments allowed for different types of comparisons (Fig. 1). Within the Landlab modeling environment we compared the outcomes of the same process equation and numerical algorithm implemented on different grid





| LEM | Modeling Environment | Grid Type | Numerical Algorithm |
|-----|----------------------|-----------|---------------------|
| CVE | CHILD | voronoi | explicit |
| LVI | Landlab | voronoi | fastscape (implicit) |
| LHI | Landlab | hexagonal | fastscape (implicit) |
| LRI | Landlab | raster | fastscape (implicit) |
| TRI | TTLEM | raster | fastscape (implicit) |
| TRT | TTLEM | raster | TVD_FVM |
| TRE | TTLEM | raster | explicit |

**Table 1.** Table of different LEMs used in this study. We use the LEM abbreviated names in column 1 to refer to the different model scenarios. The first letter in the abbreviated name stands for the modeling environment: C for CHILD; L for Landlab; and T for TTLEM. The second letter stands for the grid type: V for voronoi; H for hexagonal; and R for raster. The third letter stands for the numerical algorithm: E for explicit; I for implicit; T for TVD_FVM.

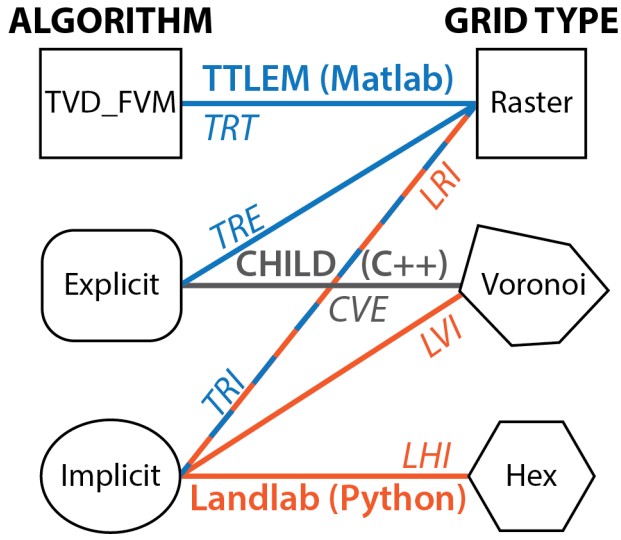

**Figure 1.** Schematic of comparison types within the experiments.

types (LRI, LVI, and LHI). Within the TTLEM modeling environment we compared the solution of the same process equation with different numerical algorithms (TRI, TRT, and TRE). Finally, among the three modeling environments we compared the

solution of the same process equation and numerical algorithm but on different grid types (CVE and TRE) and the same process equation and grid type, but different numerical algorithms (CVE and LVI).



## 4 Stream Power Process Equation and Numerical Algorithms

All of the model environments we considered use the stream power process equation to model the evolution of fluvial profiles and use or include different numerical algorithms to solve this equation. Below, we first briefly review the stream power equa-

tion and then describe the different numerical algorithms employed to solve this equation within the modeling environments.

### 4.1 Stream Power Process Equation

Within all three modeling environments, the equation controlling the change in topographic elevation at each node is,

$$\frac{dz}{dt} = U - KA^m S^n \tag{1}$$

where $z$ is node elevation; $t$ is time, and $dt$ is the model time step; $U$ is the rock uplift rate; $K$ is the erodibility parameter; $A$ is

the drainage area at a node; $S$ is the topographic slope (negative of the spatial derivative in elevation, assuming directionality is in the downslope direction) at a node; and $m$ and $n$ are positive exponents. Note that the second set of terms on the right-hand-side of Eq. 1 is the widely used stream power process equation (SPPE) that describes detachment limited fluvial incision,

$$E = KA^m S^n \tag{2}$$

where $E$ is fluvial incision. Derivations, dynamics, and limitations of the SPPE have been described in detail in numerous publications and we refer interested readers to such sources (e.g., Howard, 1994; Whipple and Tucker, 1999; Lague, 2014).

In the context of the model environments we used in our experiments, we define steady state as the condition when Eq. 1 is equal to zero, or, as said above, rock uplift rate $U$ and fluvial incision rate $E$ are equal at every node. Eq. 1 can be rearranged to solve for the analytical relationship between $S$ and $A$ at steady state,

$$S = \left(\frac{U}{K}\right)^{\frac{1}{n}} A^{\frac{-m}{n}} \tag{3}$$

Eq. 3 can be used to test whether each model experiment produces the analytical solution at steady state. As described below, this is one of the verification tests we explore in this study.

To consider the evolution of river profiles as predicted using SPPE, it is useful to define both the channel steepness $k_s$ and transformed river coordinate $\chi$. Empirically, the channel steepness was first described by Flint (1974), relating channel slope

$S$ and drainage area $A$ and considering the concavity index $\theta$ of a profile where,

$$S = k_s A^{-\theta}. \tag{4}$$

When erosion and rock uplift rate and erodibility are uniform, Eq. 4 can be recast in terms of Eqs. 2 and 3 such that,

$$k_s = \left(\frac{U}{K}\right)^{\frac{1}{n}} \tag{5}$$





and

$$\theta = \frac{m}{n}.\qquad(6)$$

Thus, on a log-log plot of $A$ and $S$ and considering Eqs. 3, 5, and 6, the steady-state form of a channel should produce a straight line with a slope of $-\theta$ and a y-intercept of $k_s$. In analyses of real channel profiles, such slope-area data is often noisy so it is useful to consider an alternative form of the same data. Specifically, it has become common to transform river profile distances $x$ using the $\chi$ integral transform proposed by Perron and Royden (2013),

$$\chi = \int_{x_b}^{x} \left( \frac{A_0}{A(x')} \right)^{\theta} dx'\qquad(7)$$

where $x_b$ is the x coordinate of base level of the profile, $x$ is the x coordinate of the channel head, and $A_0$ is a reference drainage area. At steady state and if using an appropriate value for $\theta$, a plot of $z$ (y axis) versus $\chi$ (x axis) for a channel profile will be a straight line. If $A_0$ is set to 1, the slope of that line will be equivalent to $k_s$. In the context of the SPPE, $\theta$ is often replaced with $\frac{m}{n}$ in Eq. 7, which holds when all parameters and variables are spatially uniform. $\chi$-$z$ relationships can thus be used in a
similar manner as slope-area data to assess the degree to which a particular simulation reproduces the analytical expectation.

## 4.2 Numerical Algorithms for Solving SPPE

We considered three different numerical algorithms to solve Eq. 1, specifically an explicit finite difference method (FDM), an implicit FDM, and a total variation diminishing finite volume method (TVD_FVM).

### 4.2.1 Explicit FDM

The explicit FDM is widely used and easy to implement. The explicit FDM discretizes time $t$ and space $x, z$ such that,

$$x_i = x_0 + i * dx, i = 0, 1, ..., I\qquad(8)$$
$$t_k = t_0 + k * dt, k = 0, 1, ..., K\qquad(9)$$

Using a common upwind scheme, the solution to the differential Eq. 2 is computed as,

$$\frac{z(x_i)^{k+1} - z(x_i)^k}{dt} = K A(x_i)^m \left( \frac{z(x_{i+1})^k - z(x_i)^k}{dx} \right)^n\qquad(10)$$

As described in more detail later, when using an explicit FDM, considerations must given to choices of both $dx$ and $dt$ to maintain numerical stability of the solution.

### 4.2.2 Implicit FDM

The common form of the implicit FDM for solving Eq. 2 was developed by Braun and Willett (2013), which is sometimes referred to as the *Fastscape* algorithm. This implicit FDM uses a similar discretization of both time and space as the explicit



FDM, i.e., Eq. 8. The primary difference between the implicit and explicit FDM is that the implicit solution is forward in both time and space. Specifically, the solution to Eq. 2 becomes,

$$\frac{z(x_i)^{k+1} - z(x_i)^k}{dt} = KA(x_i)^m \left( \frac{z(x_{i+1})^{k+1} - z(x_i)^{k+1}}{dx} \right)^n \tag{11}$$

Efficient solution of the implicit FDM relies on having a fixed (or known) base level for all modeled profiles at a given time step and a hierarchical list of *givers* and *receivers*, i.e., nodes that flow into each other, to overcome the challenge of having two

unknowns, specifically, $z(x_{i+1})^{k+1}$ and $z(x_i)^{k+1}$. In detail, implementing the implicit FDM is different depending on the value of $n$. If $n = 1$, then Eq. 11 can be solved explicitly, but if $n \neq 1$, Eq. 11 must be solved iteratively, but the solution converges quickly. In detail, the two model environments that we test and which use the implicit FDM, i.e., TTLEM and Landlab, find the roots of this equation differently with the former using the Newton-Raphson method and the latter using Brent's method. As our experiments are limited to $n = 1$, these differences are not directly tested here. While more complicated than the explicit

FDM, the potential benefit of the implicit FDM is that it is unconditionally numerically stable and thus in theory allows for longer time steps and shorter run times for equivalent model durations. We consider this point in the discussion of our results. For details of implementation of the implicit FDM and associated requirements, we refer readers to Braun and Willett (2013).

### 4.2.3 TVD_FVM

The TVD_FVM solution to Eq. 2 was developed by Campforts and Govers (2015) with the specific goal of minimizing nu-

merical diffusion around knickpoints within modelled fluvial profiles. There are two different solutions depending on whether $n = 1$ or $n \neq 1$, we focused on the former since our experiments were all run with $n = 1$. The TVD_FVM solution for $n = 1$ recasts Eq. 2 as a scalar conservation law,

$$\frac{dz}{dt} + \frac{df(z)}{dx} = 0 \tag{12}$$

$$f(z) = -KA(x_i)z \tag{13}$$

where the second equation represents a flux function for the conserved variable $z$. In this solution, Eq. 12 is assumed to evolve over a constant volume in both time and space where the spatial domain is discretized into finite volumes centered at $x_i$ and with a cell size of $dx$,

$$dx = \left[ x_{i+\frac{1}{2}} - x_{i-\frac{1}{2}} \right] \tag{14}$$

Eq. 12 is integrated and rearranged to derive,

$$z(x_i)^{k+1} = z(x_i)^k + \frac{dt}{dx} \left[ f_{i-\frac{1}{2}} - f_{i+\frac{1}{2}} \right] \tag{15}$$

where $z(x_i)^{k+1}$ is the elevation at a discretized gridpoint representing the average value of $z$ at the next time step and at grid position $i$ and the quantity in brackets represents the numerical approximation of the flux from Eq. 12. Other details of how the TVD_FVM algorithm works and additional steps to the solution are described in the supplementary materials of Campforts and Govers (2015), to which we refer interested readers. Of importance, the TVD_FVM is subject to similar stability criteria

as the explicit FDM.





## 4.3 Other differences between model environments

While we focus on the differences in grid type and the numerical method to solve the SPPE among the model environments we compare, this is not meant to indicate that all other aspects of these models are directly comparable. Some of the other important differences between models stem from variability in the methods for (1) calculating drainage area, (2) flow routing, and (3) identifying and filling "pits" within fluvial profiles, among other details. All of these differences have the potential to change the resulting river network geometry and landscape structure, even when beginning from the same initial condition, with the same grid type, and the same numerical solution to the SPPE. In this effort, we do not consider these aspects in detail, but highlight that future inter-model comparisons should think more broadly about differences among modeling environments.

## 5 Experimental set-up

We ran two experiments, each of which required multiple simulations with each participating model. The first experiment used all the model environments to evolve landscapes to steady state and compared steady-state morphology among the LEMs. The landscapes have the same uniform rock uplift rate, and we broadly consider that steady state is achieved when the fluvial incision rate is equal to the rock uplift rate everywhere on the landscape. For the steady-state experiments, we did not formally quantify steady state, but instead ran the models for an arbitrary long time (i.e., 100 Myr). The second experiment perturbed the previously-generated steady-state landscape by uniformly increasing the rock uplift rate, similar to previous modeling studies (e.g., Rosenbloom and Anderson, 1994; Whipple and Tucker, 1999, 2002; Gasparini et al., 2007; Attal et al., 2011). We compare the landscape morphology after 200,000 years of evolution. Further, we more formally quantified steady state and compared the time to a new steady state after perturbation.

As described below, we kept as much constant among the simulations with different LEMs. However, we did not do anything to change the models from "off the shelf". In other words, we used the modeling environments without changing any of the internal code that implements the numerical algorithms. Where appropriate, we did change free parameters within the different modelling environments to try to assure that their behavior was as comparable as possible. There are some hard-wired differences among the codes that are not user-configurable and are not related to grid types and numerical algorithms. We found these differences led to some differences in the solutions, and this will be discussed later.

### 5.1 Steady-state experiment

These simulations started from a surface with elevation values randomly chosen between 0.0 and 1.0 m. The random elevation initial surface was used because it creates more realistic looking drainage networks. The raster grids had a grid of 200 by 200 nodes, and the spacing between nodes in the x and y direction was 100 m. All the simulations using a raster grid in different models used the same exact initial topographic surface (LRI, TRI, TRT, and TRE). The numerical experiments that use a voronoi grid have an average node spacing of ≈ 100 m, but the spacing varied (in a regular way) to define the grid. The initial





condition was a similar noisy surface as used with the raster grids, and the same exact initial surface is used in the CVE and LVI simulations. Regardless of the type of grid, all the grids were $\approx$ 20 km by $\approx$ 20 km with $\approx$ 100 m resolution.

The boundary conditions used in all simulations were the same. All of the nodes on the perimeter of grid were open boundaries where water can exit, but not enter, the grid. The elevation of the perimeter boundary nodes was fixed at zero meters

and did not change during the experiments. In other words, the perimeter nodes were not uplifted but the rest of the grid was uplifted. $U$ and $K$ were spatially uniform and set at 1e-4 m/yr and 5e-6 yr$^{-1}$, respectively, in all the steady-state simulations. $m$ and $n$ were also spatially uniform and set at 0.5 and 1, respectively, in all the steady-state simulations, corresponding to a $\theta$ value of 0.5 given the uniformity of all variables and parameters.

We explored how the time step value, $dt$ in Eqs. 1, 10, 11, and 15, impacts whether the steady-state landscapes matches the

analytical solution. Each model experiment should be stable, and produce the correct analytical solution, when the time step satisfies the Courant–Friedrichs–Lewy condition:

$$C_{max} > \frac{vdt}{dx} \tag{16}$$

where $C_{max}$ is the Courant number, $\approx 1.0$ in stable conditions; $v$ is the speed that an erosional wave will move through the network, and $dx$ is the spacing between nodes. When using the stream power process equation with $n = 1$, $v$ is approximated

as

$$v \approx KA^m. \tag{17}$$

To calculate the maximum time step for a stable Courant condition ($dt$, Eq. 16), we needed to know the largest drainage area in the modeled network to estimate the fastest wave speed (Eq. 17). Because we started with a noisy surface and drainages evolved out of the noise, the area of the largest watershed was unknown until the landscape reached steady state. However, we

had to choose a stable $dt$ before the landscape evolved. Therefore we estimated the size of the largest drainage. We assumed the maximum drainage area will be one-fourth of the total area of the grid, or 49 km$^2$. This value was intended as an overestimation, ensuring that we calculated a stable time step with Eqs. 16 and 17.

Based on the parameter values that we used in Eq. 1 and the effective version of this equation within the different numerical algorithms, we estimated that $dt$ =2,500 yrs should satisfy Eq. 16 and result in stable model experiments. We explored how the

simulated landscape changed by changing $dt$. We ran four simulations with each model, reducing $dt$ to 250 yrs and increasing $dt$ to 25,000 and 100,000 yrs, or approximately 10 and 40 times greater than the stable condition, respectively.

## 5.2 Transient experiment

In these simulations we modeled how a steady-state landscape evolves in response to a uniform increase in rock uplift. Here we illustrate the behavior of largest channel profile to represent the transient behavior. We explored the variation in response

with time step $dt$ size. All LEMs that operate on raster grids used the same initial topography. We used the topography produced using the Landlab modeling environment with $dt$=2,500 yrs from the steady-state model simulations. In the model simulations using a voronoi grid, we used the topography produced from the steady-state CHILD model simulation with



$dt$=2,500 yrs. Finally, for the Landlab LEMs using a hexagonal grid, we use the Landlab steady-state topography produced using the hexagonal grid and $dt$=2,500 yrs.

Our transient model simulations increased rock uplift rate uniformly across the grid by a factor of five, to $U$=5e-4 m/yr. We did not change anything besides the rock uplift rate between our steady-state and transient simulations. We used the same four time step values used in the steady-state simulations and ran four simulations for each participating LEM. We compared the profile of the river channel with the largest drainage area after 200,000 years. We also compared the time to reach a new steady state following the perturbation. We explored how the time to steady state differs depending on the metric used for calculating

steady state and the time step $dt$.

## 6    Comparison of steady-state results

### 6.1    Raster LEM simulations

Here we illustrate the results of four LEMS (LRI, TRI, TRT, TRE) using raster grids in the Landlab and TTLEM modeling environments. All the simulations start with the same exact initial condition and used a 250 yr time step, the smallest time

step used in this study. Therefore we expect none of the variation in observed model behavior to be attributable to numerical instability. The same parameters and boundary conditions are used. The three TTLEM LEMs use the explicit FDM, implicit FDM, and the TVD_FVM to solve the SPPE, and the Landlab LEM uses the implicit FDM.

Because all these landscapes evolved from the same initial random topography and use the same time step, one might expect the details of the network structure to be identical. However, there are differences in the network organization among all the

steady-state landscapes produced with these four raster LEMs, as illustrated by the path of the largest channel across the landscape (Fig. 2). In all four cases the largest channels drain to the west of the range, but there are subtle differences in the main channel path and larger differences in the topology of the main drainage divides. These differences are highlighted in elevation difference plots, using TTLEM implicit as the base comparison case (Fig. 3. Interestingly the TTLEM implicit and explicit models produce the most similar landscapes, when measured as the sum of absolute total difference in elevation values.

Even though the TTLEM and Landlab implicit LEMs (TRI and LRI) models use the same numerical algorithm, the topographic organization differs. This could be due the details of the algorithm implementation in each modeling environment. This could also be due to details in the flow routing algorithm and how sinks (local lows) are managed. The latter is generally more likely given the greater degree of similarity in gross drainage network and divides structure between the different TTLEM runs, which all share the same flow routing algorithm and methodology for dealing with sinks (Fig. 2).

We compared the modeled slope and area data with the predicted analytical expression (Eq. 3) in Fig. 4. Despite differences in network organization, simulations from three of the four LEMs (TRI, TRE, LRI) created landscapes that exactly reproduced the slope-area relationship. The TRT model steady-state slope-area output have some scatter around the predicted analytical solution. This is not enough scatter that it is noticeable in the channel profile or chi-elevation plots (not shown here).

We suggest two takeaways from these simulations. First, be cautious when interpreting the details of planform morphology

and flow organization of either rivers or divides produced from LEMs. Even in deterministic models with uniform parameters

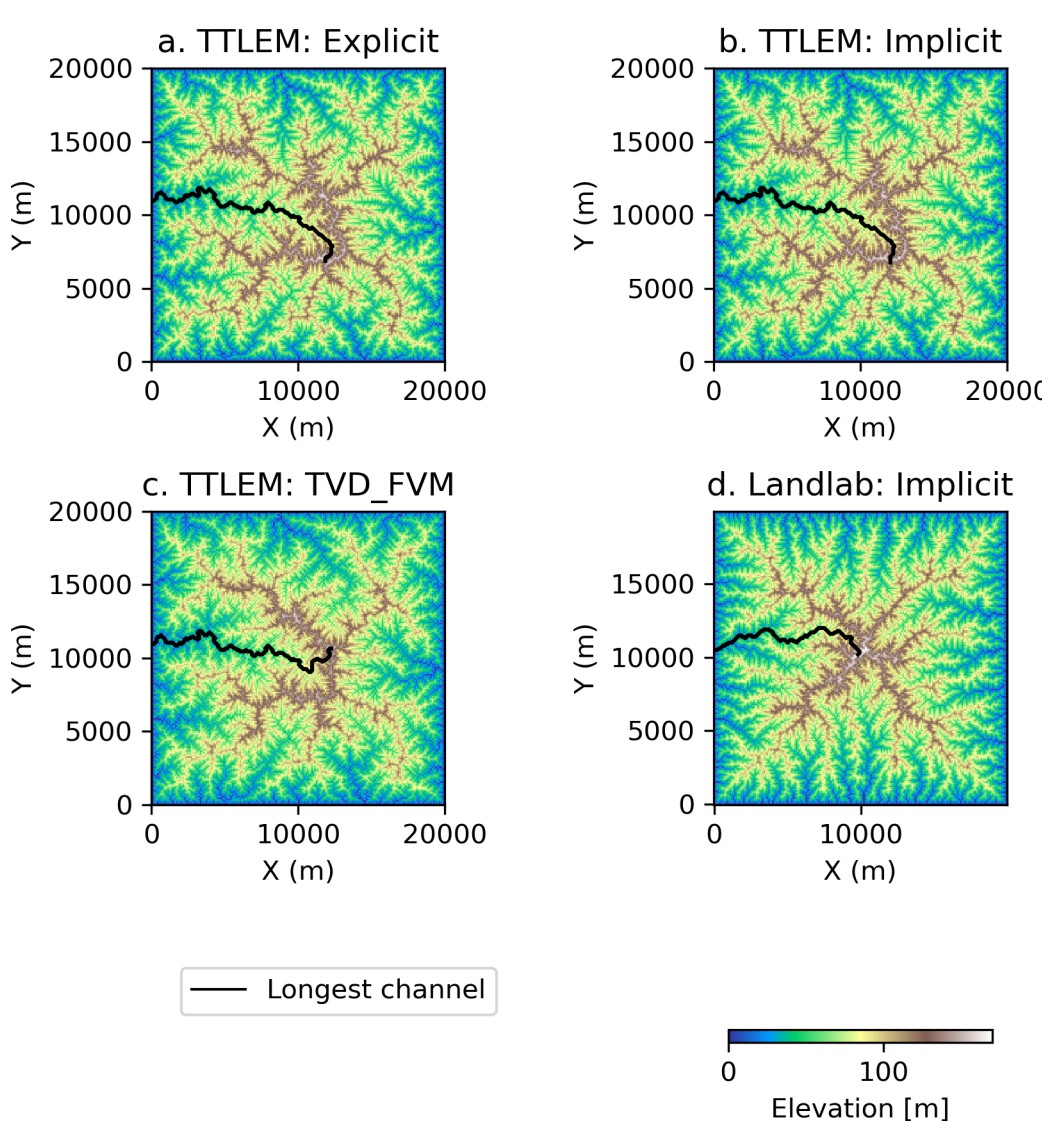

**Figure 2.** Steady-state topography of the four computer models that use a raster grid.

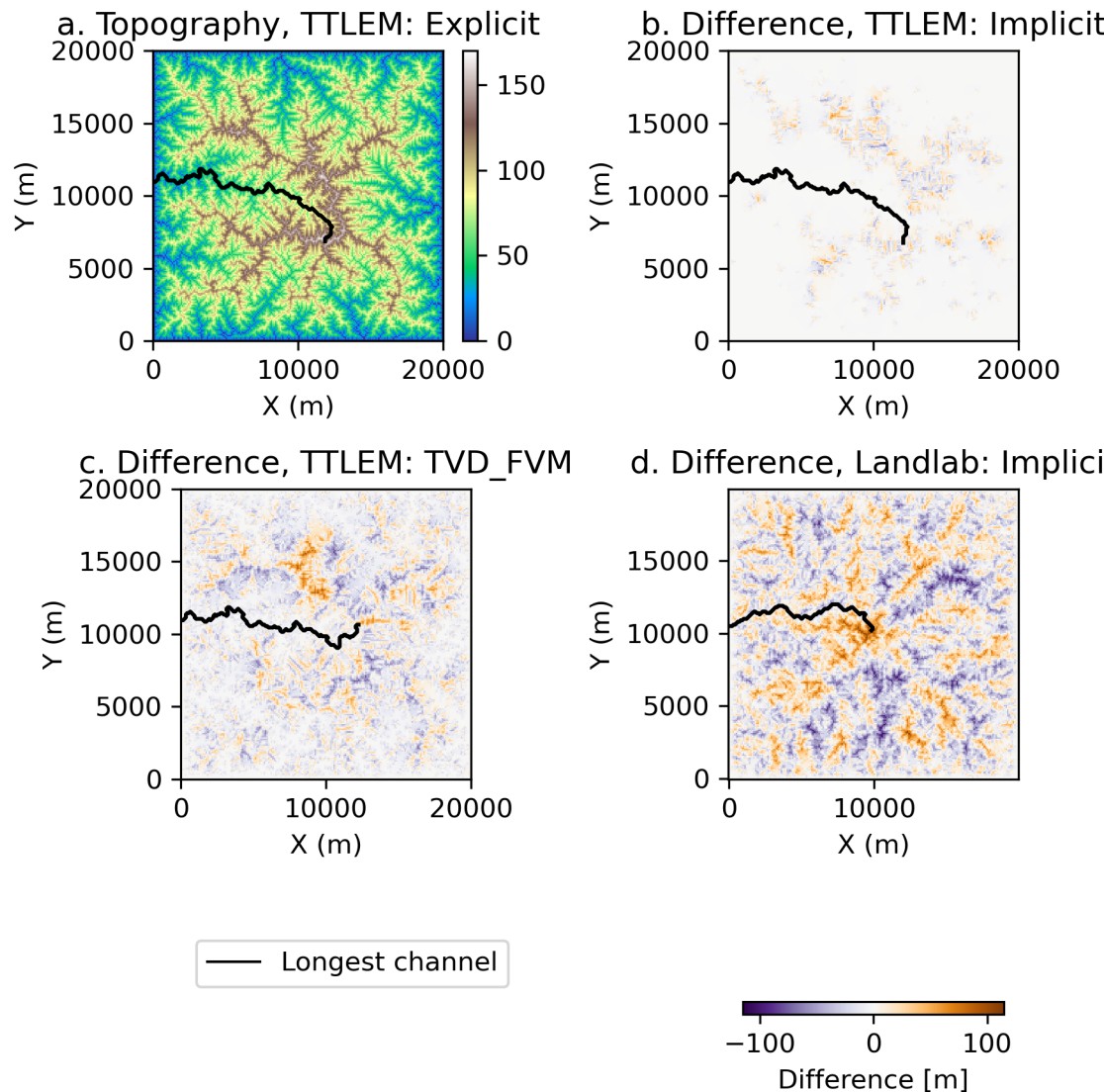

**Figure 3.** Steady-state topography produced using the TRE model (a) and the difference between (a) and the TRI steady-state topography (b), the TRT steady-state topography (c), and the LRI steady-state topography (d). A time step of 250 years was used to produce all the topographies used for this figure.

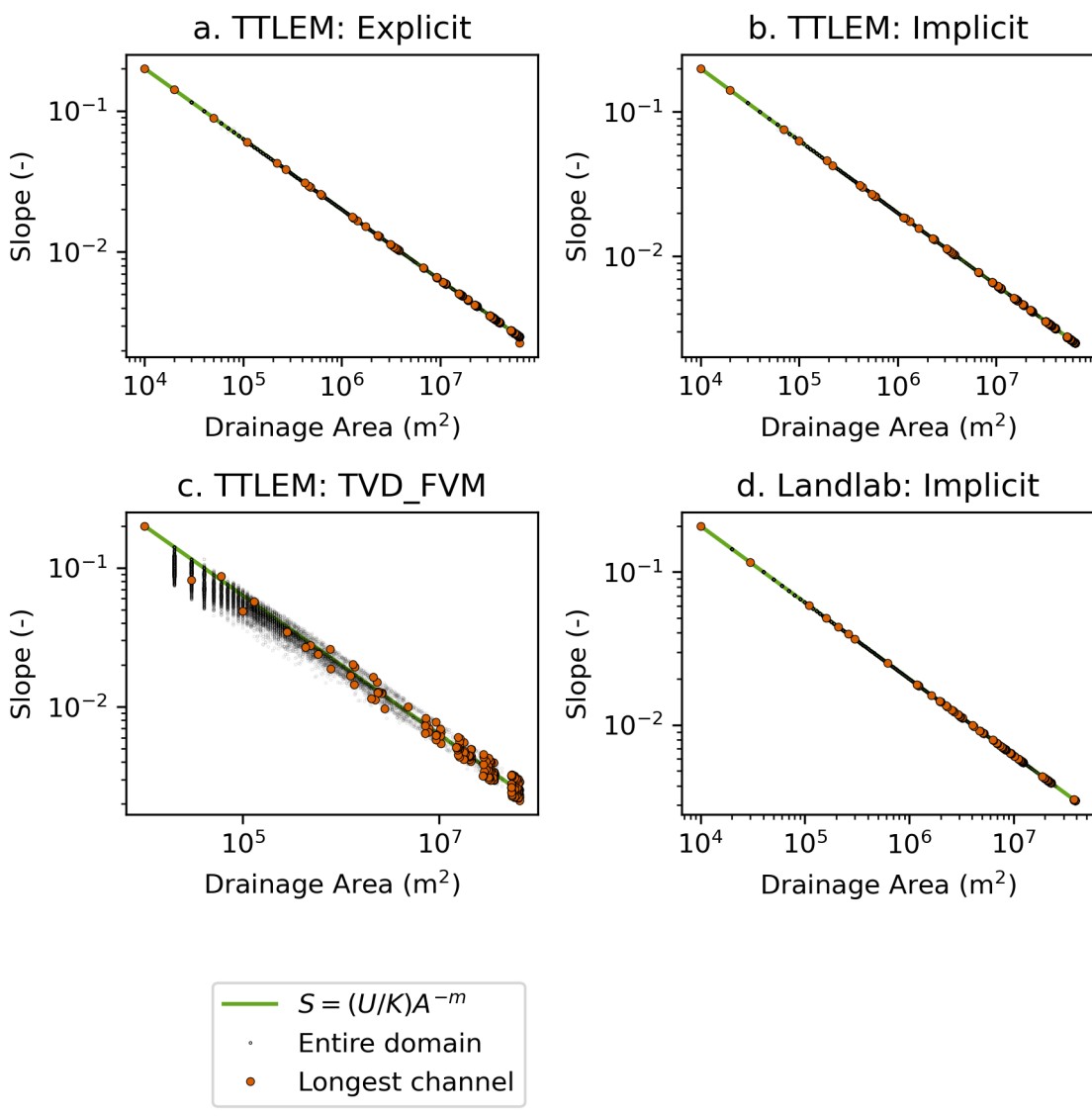

**Figure 4.** Steady-state slope-area data from the four computer models that use a raster grid. A time step of 250 years was used.





and forcings, the details of the network organization can vary solely based on the numerical algorithm and the details of how the algorithm is coded. Second, all the LEMs reproduced the predicted analytical relationship between slope and drainage area. However, some are more accurate than others. When interpreting model results, scatter that is a result of a numerical algorithm should not be misinterpreted as indicative of geomorphic processes. We suggest that the first step of any model verification study should be a test of whether the predicted analytical solution(s) is reproduced. Further, the community should agree on the accepted degree of model accuracy in reproducing the analytical solution.

## 6.2 Variable time step simulations

Here we illustrate the impact of changing the time step $dt$ between multiple simulations using the same LEM. We used the CHILD model (CVE) because the impact of increasing the time step was very clear in the output. CHILD uses an explicit FDM solution, i.e., Eq. 10. As such, it does not remain stable at time steps larger than the Courant condition (eq. 16), here estimated to be between 2,500 and 25,000 yrs.

The two landscapes produced using time steps less than the Courant condition illustrate stable behavior (Fig. 5). The analytical slope-area relationship is exactly reproduced. There are some differences between the planform morphology in the steady-state landscapes produced with the two smallest time steps, despite starting from the same exact initial condition and using the same boundary conditions. However these differences are barely detectable from the topographic maps.

The two landscapes produced using time steps greater than the Courant condition illustrate various symptoms of numerical instability. In the case of a $dt$ of 25,000 yr, the instability is not obvious from the elevation map alone. However, the scatter in the slope-area relationship illustrates that the analytical relationship is not exactly produced in the largest drainage-area reaches of the main channel (Fig. 5e). The steady-state solution is produced at smaller drainage-area reaches. This makes sense given Eq. 16, which predicts that the stable time step increases with drainage area. An even larger $dt$ of 100,000 yr has obvious instability, even in the elevation map. Scatter in the slope-area relationship is evident at smaller drainage area reaches than in the 2,500 yr experiment.

The TTLEM raster explicit LEM (TRE) has the same sensitivity (results not shown). That is, for time steps of 25,000 and 100,000 years, there was mismatch between the TRE LEM slope-area relationship and the analytical solution. This mismatch occurs in smaller drainage area reaches with a larger time step. This behavior is indicative of the numerical algorithm, and not of the grid type, as supported by the comparison between TRE and CVE. We return to the consideration of time step in the transient experiments.

The Landlab voronoi implicit (LVI) results do not show the same instability with time step as the CHILD model (CVE) does (Fig. 6). This illustrates that the instability is not an artifact of a voronoi grid. Fig. 6 also illustrates the stability of the fastscape implicit algorithm. The steady-state analytical solution is produced even when the time step is greater than that predicted from the Courant condition. Finally, the LVI model results also illustrate the change in the details of flow organization with time step.

We have three take-aways from the variable time step steady-state numerical experiments. First, the details of flow organization can vary with model time step, even when the same LEM is used with the same flow routing and pit-filling algorithms.

Earth **Surface**
**Dynamics**
Discussions

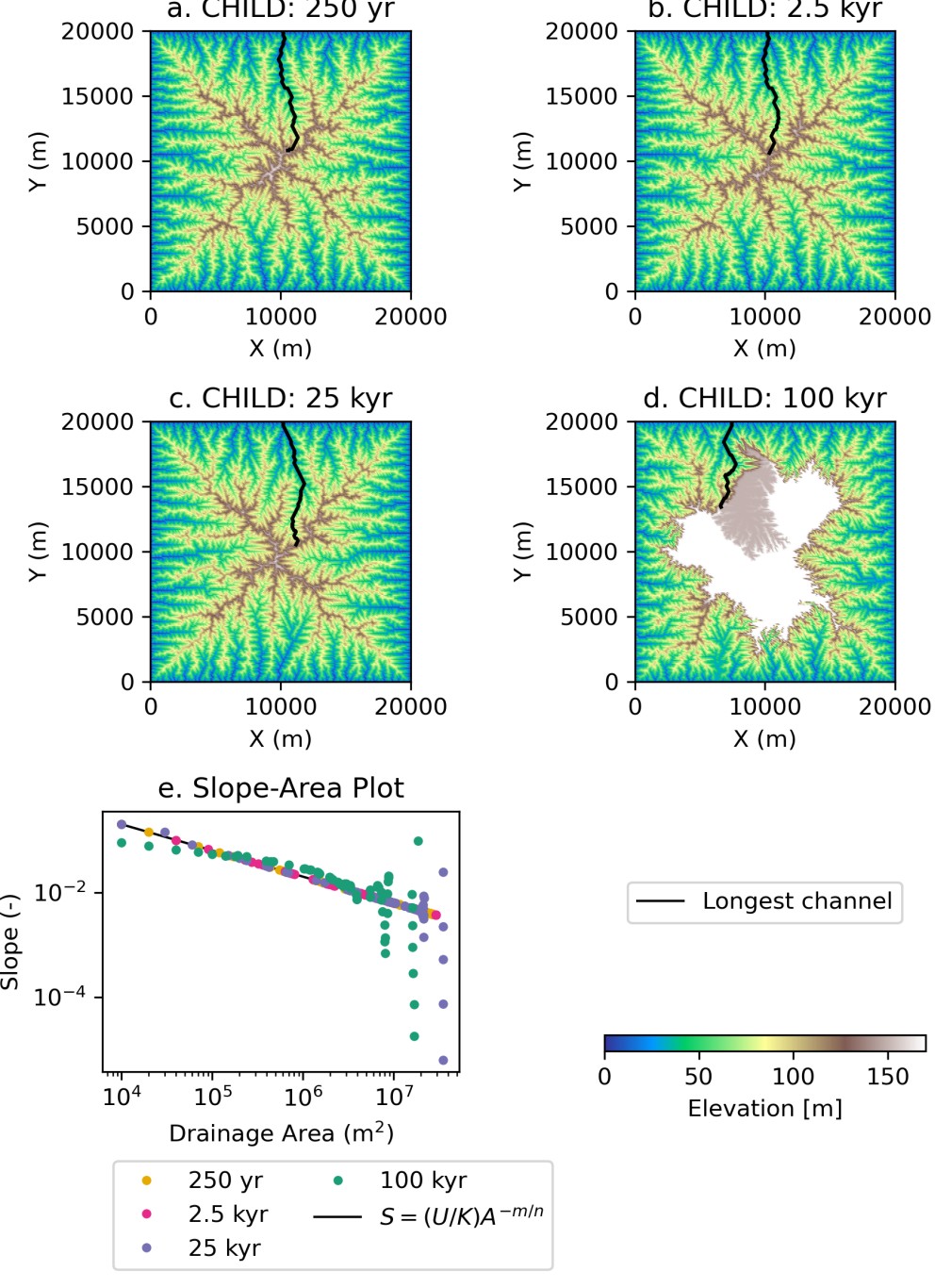

**Figure 5.** Topography of the entire model domain (a, b, c, d) and slope-area plot (e) of the largest channel from the CHILD steady-state simulations using different time steps.







**Figure 6.** Topography of the entire model domain (a, b, c, d) and slope-area plot (e) of the largest channel from the Landlab voronoi (LVI)
steady-state simulations using different time steps.



Second, model instability is not always obvious from a topographic map. If a LEM does not exactly reproduce an analytical solution, it may be an indicator of numerical instability. Third, the stability of the LEM depends on the time step and numerical algorithm. We did not find the stability of a LEM to be a function of the grid type.

## 7  Comparison of transient results

### 7.1  Morphology of transient landscapes

Here we illustrate the transient behavior of a landscape as it adjusts to an increase in rock uplift rate (5e-4 m/yr) from steady state with a low rock uplift rate (1e-4 m/yr). We used three TTLEM LEMs (TRI, TRT, TRE) because they allowed us to illustrate the behavior of three different numerical algorithms and their sensitivity to time step during a perturbation while keeping other details (e.g., grid type, flow routing, etc.) the same.

The transient morphology of the largest channel after 200,000 years of evolution using the TRI, TRT, and TRE LEMs is
shown in Figure 7. The solution differed among the algorithms. None of the algorithms produced the exact solution above a time step of 2,500 yrs. We predicted that 250 and 2,500 yrs should both be stable time steps, and there are only slight differences in the solutions produced with these two time steps. The model inaccuracies produced with all of the numerical algorithms with a time step of 25,000 or 100,000 yrs are easily apparent in comparison with the solutions using smaller time steps. Below we discuss the details of the results produced with the different numerical solutions.

The implicit numerical solution (left column of figure 7) smoothed the shape of the knickpoint when the time step did not satisfy the Courant condition (Eq. 16). In the simulations with a time step of 250 or 2,500 yrs, there was a sharp transition from the upstream, more slowly eroding part of the channel above 100 m elevation and the downstream part of the channel that has already responded to increase in rock uplift. The $\chi$-$z$ plots (middle row of figure 7) show the divergence from the expected steady-state solutions (dashed and dotted lines) in the two simulations with the largest time steps. In the channel profile (top
row of figure 7) this appears as a more rounded knickzone, as the transition from one erosion rate to another is spread above and below 100 m. The spreading of the knickzone with time step is visible in the slope-area plots (bottom row of figure 7) as the experiment with the longest time step illustrates that points with smaller drainage area have started to adjust to the change in rock uplift. This smoothing of the knickpoint does not impact the total relief in the channel.

In contrast to the implicit solution, when the time step is greater than 2,500 years both the TVD_FVM (middle column of
figure 7) and explicit FDM (right column of figure 7) numerical algorithms generated behavior commonly associated with numerical instability. The channel profiles for both numerical algorithms had a stair-step appearance below the elevation of the knickpoint (100 m) with the 25,000 and 100,000 yr time steps. When the time step was 100,000 yrs, there were only two time steps modeled in these examples. Given the rock uplift rate, each time step uplifts the landscape by 50 meters. The impact of these two large uplift *events* are clearly visible in the channel profiles and elevation-chi plots with both the TVD_FVM and
explicit numerical algorithms. If you look very carefully, the impact of the 12.5 meter uplift *events* with the 25,000 yr time step is also visible in the lower portions of the profiles. Both algorithms smoothed the smaller profile steps (made with a $dt$ of 25,000 yrs) as they move up through the landscape.





**Figure 7.** Elevation, $\chi$ profile, and slope area data of main channel after 200,000 model years of evolution. All simulations used the TTLEM modeling environment.



The CHILD LEM (CVE) uses an explicit algorithm, and the results from the same numerical experiments with CVE match those produced with TRE (results not shown, only performed with time step $\leq 25,000$ years). We note that CHILD and TTLEM

have internal stability criteria which are meant to reduce numerical instabilities like those illustrated in figure 7. Usually these internal stability corrections in a model reduce the time step over which a process is calculated. However, these stability corrections only apply in the fluvial process calculations in both models and do not impact the time step over which rock uplift is applied. Thus, even with stability correction in both CHILD and TTLEM, a long time step creates a large cliff on the boundary of the model when rock uplift is added to the landscape.

## 7.2   Time to steady state

Here we compare the time to reach a new steady state following the perturbation used above. We illustrate the results using the three Landlab LEMs (LVI, LHI, LRI) and the three TTLEM LEMs (TRI, TRT, TRE) using different time steps. Thus we are able to document whether grid type, numerical algorithm, and/or time step impact time to steady state.

We illustrate the time series of four different metrics whose value or temporal behavior could be indicative of steady state.

The four metrics are the temporal change in maximum elevation, mean elevation, maximum location elevation, and sediment flux eroded from the entire landscape. We refer to these collectively as the steady-state metrics. In all cases the steady-state metrics are calculated over temporal difference 100,000 yrs, which was set by the longest time step of the simulations being compared. Therefore the first time that any metric was calculated is at 100,000 yrs into a transient simulation. This was done to ensure equivalent comparisons between the simulations as some degree of differences in time to steady state would be expected

if we calculated these metrics over different time intervals between simulations.

The temporal change in maximum elevation, $\Delta z_{max}^t$ was calculated as,

$$\Delta z_{max}^t = |max\left(z_i^t\right) - max\left(z_i^{t-100,000}\right)|, \tag{18}$$

where $max\left(z_i^t\right)$ is the maximum grid elevation within the domain at time $t$; and $max\left(z_i^{t-100,000}\right)$ is the maximum grid elevation within the domain at time $t - 100,000$.

The temporal change in mean elevation, $\Delta z_{mean}^t$ was calculated as,

$$\Delta z_{mean}^t = |mean\left(z_i^t\right) - mean\left(z_i^{t-100,000}\right)|, \tag{19}$$

where $mean\left(z_i^t\right)$ is the mean grid elevation within the domain at time $t$; and $mean\left(z_i^{t-100,000}\right)$ is the mean grid elevation within the domain at time $t - 100,000$.

The temporal change in maximum local elevation, $\Delta z_{max(loc)}^t$ was calculated as,

$$\Delta z_{max(loc)}^t = max\left(|z_i^t - z_i^{t-100,000}|\right). \tag{20}$$

The difference between equations 20 and 18 is that the former finds the difference between the maximum elevation of the entire domain at two different times, whereas the latter finds the difference in elevation at every node in the landscape between two different time steps and uses the maximum of those differences.



Finally, the temporal change in sediment flux, $\Delta Qs^t$, was calculated as,

$$\Delta Qs^t = \sum Qs_i^t - \sum Qs_i^{t-100,000}, \tag{21}$$

where $\sum Qs_i^t$ is the summation of the erosion rate at every node on the landscape at time $t$; and $\sum Qs_i^{t-100,000}$ is the summation of the erosion rate at every node on the landscape at time $t - 100,000$. The models do not track the sediment flux explicitly when using the SPPE, which is why we use the local erosion rate as a proxy for sediment flux.

As a landscape evolves towards steady state, all of the steady-state metrics should approach zero. When exactly a new steady state is reached could be defined as when the metric value passes below a predetermined threshold. The strictest definition would be when a metric reaches zero. We note that differences in the accuracy of calculations in different modeling environments could lead to some of the metrics never reaching zero. Also, the time at which a particular metric appears to reach zero will depend on the floating point precision of the programming environment in which the metric is calculated. One could also use the rate of change in the value of a steady-state metric as a criteria for reaching steady state. As the intent of this paper was not to determine the best metrics or methods for LEM comparison, we do not settle the discussion. Rather we present the results for discussion among the community.

Each column of Figure 8 shows the time series of one of the four steady-state metrics. The top row of Figure 8 illustrates the results for TRI and LRI LEMs, which have the same grid type and numerical algorithm. Note that if one were to choose a threshold value for any of the elevation steady-state metrics, say $10^{-5}$ m, the estimated time to steady state varied by as much as 10 my for the same model environment depending on the time step. When comparing between the two model environments and the different time step values, and using the same threshold, the time to steady state varied by almost 20 my. A similar conclusion would also be reached if one chose a threshold value less than $10^{-5}$ m/yr with the sediment flux steady-state metric.

In contrast, if one were to use the time at which a metric asymptotes to a steady value, the estimated times to steady state would be closer for a given metric. This is regardless of the time step or the LEM in the simulations with a raster grid. However, the value that a given metric asymptotes to differs with time step in some cases (e.g. the sediment flux steady-state metric in Landlab simulations with different time steps).

The approach to steady state as gauged by the time series of the four different steady-state metrics in the transient Landlab LEMs using hexagonal and Voronoi grids (LHI, LVI) was similar to that of the TRI and LRI LEMS. In the hexagonal grid LEMs, when comparing among the time steps, some of the metrics could lead to ≈ 5 my estimated differences in time to steady state if using a threshold value. The time to steady state when calculated using a threshold value also varied among the Voronoi simulations with different time steps. However, there was less variation with the Voronoi simulations than there was with the hexagonal simulations.

The steady-state metrics from the TTLEM TRT and TRE experiments showed similar behavior as discussed above, but there are some caveats. First, these numerical algorithms are not stable with larger time steps. Thus, the impact of time step on the steady-state metric values was not large, given we only illustrate the time steps used satisfied Eq. 16. We are unsure why there were temporal oscillation in three of the steady-state metrics using the TRT model. Possibly this is related to the fact that the TRT model never converges to the exact analytical solution (Fig. 4).



Earth **Surface**
**Dynamics**
Discussions

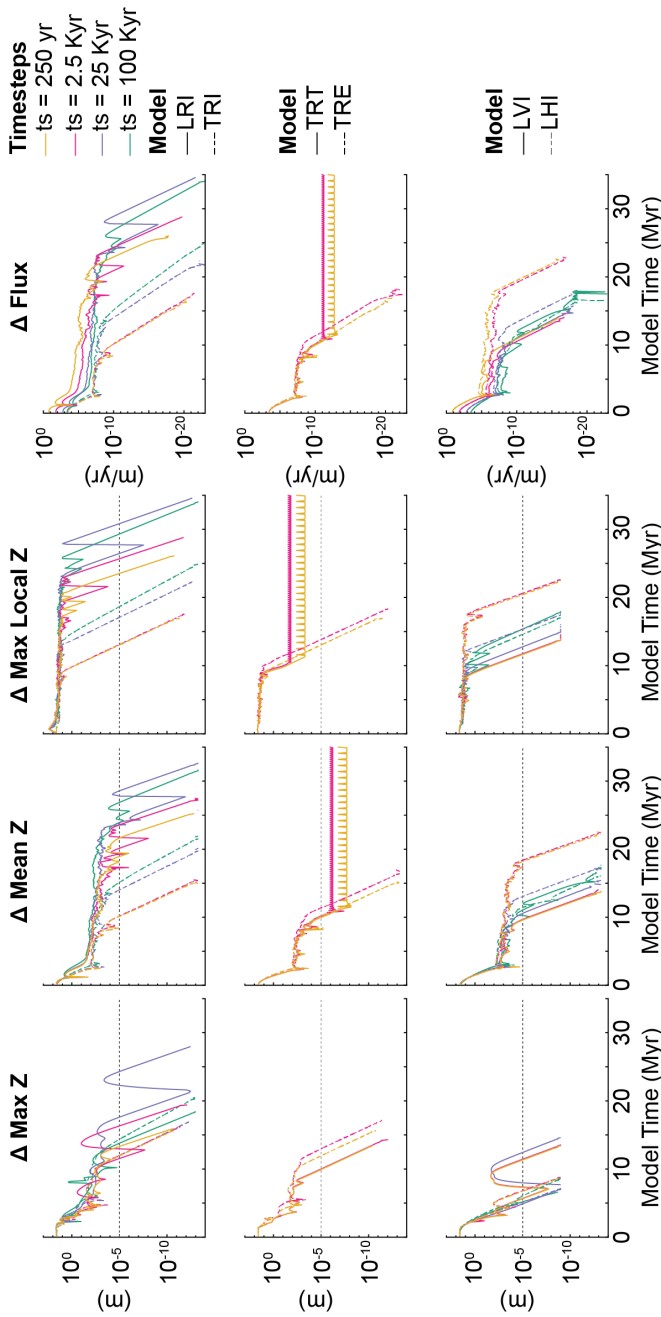

**Figure 8.** Time series of four different steady-state metrics that can be used for evaluating when a landscape has reached steady state. All metrics are calculated following an increase in rock uplift.





The biggest take-away from this section is that time-to-steady-state metrics are likely not comparable among modeling studies. Here we used the same network for all of the raster transient simulations, so we eliminated network variability as a
cause for differences in time to steady state. We showed that the metric and value for determining steady state matter, along with the time step, grid type, numerical model, and numerical algorithm.

## 8   Summary, Implications, and Conclusions

### 8.1   Stability, accuracy, verification, and the domain of applicability

A stable model avoids some of the tell-tale saw-tooth behavior of an unstable model (e.g. Fig. 5), but a stable model is not
always accurate. We can see this in the transient behavior of the models that use the implicit solution, i.e. Fastscape numerical algorithm with long time steps. The shape of the knickpoint is rounded when compared to the LEM that uses the TVD_FVM numerical algorithm, which is designed to maintain the shape of a knickpoint. However, the same algorithm that is able to maintain the shape of the knickpoint has scatter in the slope-area relationship at steady state.

Whether stability or accuracy or both are important in a modeling study depends on the scientific hypothesis being tested.
What is always important is to not interpret instability or inaccuracy as being indicative of landscape processes. Model results that have clear instability have been presented in previous studies. For example, instability in fluvial processes near the outlet of a river, where drainage area is high and the boundary condition is imposed on a landscape. However, upstream of this instability, the model behaves accurately, and therefore, as long as only the stable and accurate part of the solution is used for hypothesis testing or scientific exploration, there is no problem.

We posit that the more important consideration is that the LEM community has not formally or informally defined the domain of applicability or engaged with model verification through benchmarking. Community driven definition of benchmark experiments and formal verification of models will clarify the circumstances under which a particular model is an appropriate choice for a scientific question. Furthermore, such benchmarks will demonstrate which aspects of model output are reliably interpretable from any given model (e.g., some studies may only want to interpret the relief of a landscape while others want to
interpret the transient drainage reorganization). Enumeration and implementation of benchmark experiments associated with the most commonly interpreted LEM outputs would not elucidate the domain of applicability for each model, but these experiments would likely support beginner modelers in understanding which models are appropriate to use for different applications.

Beginning LEM users must make numerous decisions. First, they must choose a modeling environment. Next they must make decisions on model initial conditions, boundary conditions, time step, and model resolution. They may also have to decide on
grid type. Adding to the complexity of these decisions is the fact that different process models may apply on different spatial scales. As discussed by (Skinner and Coulthard, 2022), some decisions on model set-up are made arbitrarily based on model efficiency. By defining a domain of applicability for different LEMs, the community could remove the guesswork from these decisions.



## 8.2 LEMs are not good at everything

Our results highlight the danger in over-interpreting time to steady state from a LEM. When a model output is time step dependent or multiple models do not converge on the same answer, even when run with time steps below the Courant condition, it suggests that interpretation of that output may not be possible or must be done with caution. We illustrated that time to steady state is not just time step dependent, but also algorithm, grid, and model environment dependent, as well as metric dependent.

Our study was not extensive. A much deeper model comparison study could reveal other LEM strengths and weaknesses.
This study only compared models using the stream power process equation. However, there are other processes shaping a landscape that may be included in different LEMs. There are also other fluvial process equations in LEMs. Most LEMs require flow routing and drainage area calculations in order to calculate fluvial transport or incision rates. Flow routing and drainage area algorithms can vary in and among modeling environments, and these details have been shown to matter (e.g., Shelef and Hilley, 2013; Adams et al., 2017; Lai and Anders, 2018). Explicitly stating what different LEMs can and cannot do is critical
for appropriate model use.

## 8.3 What (could be) next?

The surface processes community is pushing LEMs further and further. LEMs are being used for global prediction of sediment flux over 100 my (Salles et al., 2023). On the opposite timescale, Chen et al. (2021) used a LEM to explore how humans have impacted soil erosion over the Holocene. We are extremely excited about the proliferation of LEMs, and their use in exploring
diverse surface processes questions. This paper is **not** suggesting that LEMs are only for an exclusive group of users. At present though, many of the limitations, best practices, and challenges related to the use of particular modeling environments are only known to small groups of users and are communicated informally. This informal sharing of knowledge makes using LEMs inherently exclusive despite the ease of access of the models themselves. In response, we think that in order for LEMs to be used by everyone, our community deserves and requires a better, widely communicated, quantitative understanding of the
strengths, limits, and pitfalls of different LEMs. Fortunately, a large number of scientific disciplines have already established practices for model benchmarking that the LEM development and application community can learn from.

We suggest that the community of LEM developers and super users—those who have vast experience in using LEMs— should come together to design benchmark experiments that illustrate what particular LEMs can and cannot do, similar to other communities in the geosciences. These benchmarks would likely start with reproducing known analytical solutions and
then expand into more difficult targets such as transient behavior and planview drainage patterns. Where analytical solutions are not known, established methods, such as the method of manufactured solutions (Roache, 1998) provide a way forward. A possible guide for identifying and prioritizing benchmark experiments is the type of model output most commonly interpreted.

After a benchmark experiment is designed, individual models can submit their performance. The collective development of these benchmarks and individual model performance on each benchmark would result in illuminating the domain of applica-
bility for each model. That is, what are the reasonable parameters, boundary and initial conditions, grid resolutions, and time steps for which a model will provide reliable results?



Ultimately, the development of a vibrant culture of model benchmarking for LEMs will make our codes more reliable, our documentation better, and our science stronger. The possibilities grow rapidly and well beyond what was presented in this study. Yet even our highly contained exploration illustrated pitfalls that we, as experienced modelers, didn't know about. This raises the question of how some of what we presented might be interpreted by a novice modeler. No model developer wants results from their software incorrectly interpreted.

How such an effort might come to be remains an open question. Likely there would be multiple meetings and white papers written. Ideally the meeting participants would include scientists new to numerical modeling, as well as scientists who have designed and run models for decades. It would be important to include participants from diverse surface processes communities. For example, those who work on long and short time scales and those who work in diverse landscapes, e.g. glaciated, arid, and tropical. Scientists from different countries and different types of institutes should be part of the discussion. Only with broad community buy-in would such an effort have impact.

We also want to emphasize that model comparison is not solely about identifying when models do not work. It also highlights the strengths of different models and which models are best for testing different hypotheses. All modelers know the George Box saying "all models are wrong, but some models are useful". Maybe a more positive spin on the saying would be "all models are useful for something, but you have to know what that something is". It's clunkier and won't take off like Box's saying, but the nuance is important. If we explicitly test for the strengths and weakness of a model, the science using LEMs will be stronger.

*Code and data availability.* CHILD is distributed from this repository: https://github.com/childmodel/child; accessed 1 May 2023. Landlab is distributed from this repository: https://github.com/landlab/landlab; accessed 1 May 2023. TTLEM is distributed from this repository: https://github.com/wschwanghart/topotoolbox; accessed 1 May 2023. The initial grids, input files, and code for creating the figures for this manuscript are available from this repository: https://github.com/nicgaspar/LEM_comparison.

*Author contributions.* NG initiated this study and ran all the CHILD model experiments. KB ran all the Landlab model experiments and made figures for the manuscript. AF ran all the TTLEM experiments and made figures for the manuscript. NG wrote the first draft of the manuscript. All authors contributed to the design and direction of the study, interpreting the results, and refining the manuscript.

*Competing interests.* The authors have no competing interests to declare.

*Acknowledgements.* We gratefully acknowledge funding from the Tulane Oliver Fund, NSF Awards OAC-1450338 (NMG), EAR-1349375 (NMG), and EAR-1725774 (KRB). We acknowledge contributions from Nathan Lyons in the initial stages of this project. Conversations with Kelin Whipple, Brian Yanites, and Leif Karlstrom encouraged us to pursue this work. CSDMS enabled this modeling study. We also



acknowledge the impact of the global COVID-19 pandemic on this study. The first draft of this manuscript was written in January 2020, but the work was put aside for two years due to NMG being overwhelmed by the excess work and family burdens created during a pandemic.



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
