# Peer review of "Short Communication: Numerically simulated time to steady state is not a reliable measure of landscape response time"

_Earth Surface Dynamics, 2023_

## Referee Comment (RC1)

**Peer-Review of**
**"Short Communicaton: Motivation for standardizing and normalizing inter-model comparison of computational landscape evolution models"**

Anonymous referee

July 6, 2023

The manuscript "Short Communicaton: Motivation for standardizing and normalizing inter-model comparison of computational landscape evolution models" presents a call to the geomorphologic community to devise a concept for the benchmarking of numerical landscape evolution models. The authors give a short overview make why these efforts are necessary and provide a inter-model comparison of potential numerical artefacts connected to timestep length that can affect steady state computations of landscapes between three model softwares that implement a simple detachment-limited fluvial erosion law.

After reading the manuscript I am not quite sure what the authors want to accomplish overall. The manuscript appears a bit unfocused and seems to change scope midway. The introduction makes a good case for the need of benchmarking of LEM models (which I will not dispute, although my idea of what should be benchmarked seems to differ significantly), but then offers a partial introduction into the basics of numerical methods and a demonstration of issues than arise mainly by the choice of numerical schemes and by neglecting the stability criterion.
I do understand that the authors try to span a bridge here between model users and model developers by including a detailed introduction of basic numerical modelling theory. The authors comment on the need of teaching beginners the correct handling of model, but I do not think that this is a topic of scientific research and better suited for presentation in a book, where the theoretical background can be sufficiently covered and a detailed demonstration of known numerical issues can be better appreciated. It seems that the actual topic of benchmarking was completely sidetracked by the need to explain everything from scratch. Additionally, the discussion is still missing salient points in my eyes and does not even begin to treat LEM algorithms that have a big influence on the results presented here.

The concept of benchmarking that is executed here does not seem to be adressing the needs that are put forward in the introduction. The authors define the scope of their concept of benchmarking to a "known answer (that) may derive from the analytical solution to the model governing equations" (L. 21f) and the idea that "model performance on benchmark problems indicates (that) the model can reliably solve that type of problem" (L.23f).

Concerning the first statement, I expected that for benchmarking purposes as they are referenced in the introduction, model results should be compared against a "solution" that is independent of the model, for example a natural dataset or an independent analytical solution, or even against a

specific model that is considered state of the art. Deriving a prediction from the model equations itself and testing it just tells us if the model can solve its own model equations correctly, but gives us no measure how well we approximate the natural process. To be frank I expect that model developers test routinely if their model works technically before release. Unless a formerly unknown problem turns up, I do not see the benefit of revisiting these test.

Concerning the second statement, the first issue that should be adressed is which transport law comes closest to reproduce natural processes. There are different models around that include or exclude sediment transport and have different concepts to deal with how sediments are transported and deposited. I at least expected a short discussion that highlights why the detachment-limited model is considered a good candidate. If purely detachment-limited erosion turns out to be the worst approximation, comparatively small deviations that arise from different numerical implementations probably do not matter so much anymore. Concentrating on a specific transport law before being sure that it is a useful law is a bit like saddling the horse from behind.

Another issue with the tests is the focus on purely numerical issues concerning accuracy and stability that arise from the discretization scheme. These problems are not particular to LEM software or numerical models, but to every mathematical and numerical scheme that uses discretized difference quotients and iterative steps. I think it really should not be surprising that the quality of model results is compromised when i.e. the temporal resolution of the model is coarsened too much. Also I do not see the sense in demonstrating that ignoring an established stability criterion for dt leads to wrong and unusable results. To be frank, I learned about these issues in my first lecture on numerical methods and every book on the topic covers these issues and how to handle them. Most of the shown issues are vastly improved or even disappear if a reasonable dt is chosen. (I will point out a few examples in greater detail in the line comments further down to illustrate this point.) Actually, I expect that a benchmark experiment for inter-model comparison is designed in such a way that numerical inaccuracies arising from the scheme are reduced as much as it is possible and sensible, so that the actual model differences with regard to model equations and algorithms are brought out. As a model user interested in the capabilities of other models, the comparison of essentially wrong results or results obtained outside of a sensible parameter range does not help me much. Also there is a massive backlog of mathematical literature that dates back decades (The associated paper from the namesakes of the Courant-Friedrichs-Lewy criterion is from 1928!) that is basically ignored in the discussion. The convergence and accuracy of discretized schemes with regard to their analytical solutions (especially concerning the advection or wave equations that the detachment-limited model belongs to) are better understood than shown here. What I also miss in the discussion is a full comparison of the model, including the algorithms (flow routing, handling of sinks, etc.) that actually make these models "landform evolution models" and not just "advection models". That these other algorithms play quite a big role for the result of identical experiments becomes very clear in Figure 2 and 3, where the difference between results using different software is quite obvious, but the the same model software just using different numerical schemes produces very similar topographies.

The practical benefit of the tests is additionally not clear to me. What is the significance of testing the time to steady state or comparing different steady state solutions that evolved from random initial topographies if we do not have a measure which of these solutions is the "truest"? The model results differ only by divide migration, but all results are valid steady state solutions. What the study lacks here is an independent solution that gives a measure which of the LEM softwares gives a closer approximation of nature.

In conclusion, the authors essentially demonstrate that the selected models work as they are intendend, which is reassuring, but not surprising considering that the model softwares are used for quite some time. The study mainly focuses on the investigation of different numerical schemes

that solve an advection equation, but cannot add anything new here. Known issues are not adequately discussed and relevant theoretical background is often omitted. The shown results essentially confirm established mathematical theory and demonstrate known issues. Many of the differences shown here are the direct result of using a dt that is intentially unstable or too large to adequately resolve changes in the process rate. For an inter-model comparison, I expect that models are handled correctly in such a way that accuracy is optimized and on equal grounds. If this issue is remedied, I do not expect that there is much left to discuss, considering that the models are very similar and that the authors are unwilling to discuss the influence of other model algorithms. The latter is also part of the reason that I do not see a real inter-model comparison of LEM models here because most of the algorithms that affect the outcome of different models are not discussed at all. Additionally I miss an independent criterion for the quality of results. Differences are pointed out, but there is no measure that highlights one model over the others.

Based on the overall impression I recommend rejection of the manuscript.

The following section provides a few line comments that underline the reasons that led to my recommendation to reject the manuscript. It is not exhaustive, I focused on examples that underline my criticism. Additionally, since the authors expressively state that they want to initiate a discussion among the greater community, I felt motivated to add a few thoughts that arise from my own modelling experiences.

**1 Major Remarks**

**1.1 Title**

Is this a short communication? The structure of the manuscript, especially the terminology section and detailed explanation of numerical algorithms and model equations of the different models gives a strong impression of a review paper.

**1.2 Introduction**

*L.21:* The comparison against predictions stemming from the model equation just tells if the model can correctly reproduce the assumptions that are put in at the beginning, but gives no indication if the model equation can approximate the natural process. I specifically miss a discussion on expectations of how well the detachment-limited erosion law represents nature.

*L.33:* These references are maybe misleading as role models for this study, since they are concerned with benchmarking models based on different governing equations and comparison against independent data like natural datasets (historical climate data) and analogue experiments (critical taper sandbox models).

*L.51:* I am puzzled about the "statistical" nature of deviations in outcomes of the same model simulation used in different model softwares. Do the authors do not want to imply that the models have a random component? I assume that LEM simulations give reproducable results as long as no parameters are changed.

*Terminology section* This section seems more appropriate for a review paper or tutorial and might not be useful here. I understand that the authors want to bring everybody on the same page and want to open the discussion on appropriate benchmarking experiments for members of the community that are not intimate with the internals of numerical models, but I expect that

people interested in computer models have a good idea about at least some of the terms.

*Figure 1:* I have trouble with this diagram. Assuming that the colors indicate groups of models that are compared against each other, it seems that the CHILD model is not compared against anything?

*L.189:* The variable $A(x)$ is spatially uniform?

*L.195f:* The following introduction to numerical schemes seems not to be really used at a later point to explain the signatures and differences between the modeling softwares. Is it really necessary?

*L.196:* The explicit scheme is not limited to regular grids and the discretization of the raster with a constant $dx$ is not valid for the voronoi, which has a variable $dx$. I do not think this section should be expanded, but it does certainly not help if the information is incomplete with regards to all the model variations that are tested here.

*L.203:* I think the implementation of this particular implicit scheme was formerly proposed by Hergarten & Neugebauer (2001, doi 10.1103/PhysRevLett.86.2689).

*L.236f:* In Fig. 2 and 3 and thought it seems quite obvious that these differences in other model algorithms seem to be responsible for the larger part of the differences between the results of different model softwares. The authors note themselves in L. 321f that the strong similarity in river networks for the two TTLEM simulations is likely due to the same algorithms for flow routing, etc.. Considering that the TTLEM implemented the same implicit scheme from Landlab, the best explanation for the large differences there is the great importance of these other algorithms. How can the authors uphold their confidence that these other factors can be ignored? From my own experience I expect that differences in flow routing, calculation of discharge and handling of sinks will strongly affect the migration of drainage divides.

*L.290:* The merit of purposely working with a $dt$ that results in an unstable simulation eludes me. Unstable means that the results systematically drift away from the analytical solution and errors accumulate, so the results should be considered wrong and put into the bin. I am absolutely not a fan of treating instability as a kind of inaccuracy, because this implies that results are still considered to be somehow usable. Maybe I get the intent of the authors wrong here, but establishing that model results are unusable and then continuing to evaluate them sends a wrong signal here.
Another thought: 25 or 100 kyr are impratically (not to say ridicilously) large timesteps. Under changing boundary conditions, erosion rates undergo a gradual change and the time resolution must be fine enough to adequatly resolve that. Otherwise the result can be such stair stepping in the river profile as shown in Fig. 7 later in the manuscript.

*L.325:* The predicted slope-area relationship in steady state is essentially derived directly from the model equation for the SPPE model. The TRT model, however, essentially adjusts how the erosion rate is calculated to counteract numerical diffusion that smears knickpoints. Doesn't that mean that the analytical model prediction must also be adjusted here to make a comparison on equal grounds? Alternatively, are slopes and catchment sizes recalculated/averaged according to their usage in the model equation? If not, that might at least explain the scatter. I wonder about the flattening, but my first guess would be that this is the result of adjusting the calculation and not numerical inaccuracy. If the authors actually want to imply that the predicted slope area relationship is somehow reprasentative for natural systems, I definitely missed the discussion (and

I would probably disagree, remembering natural datasets where the relationship breaks down at small drainage areas).

*L. 329f:* It is not surprising that numerical models with slightly different algorithms produce slightly different results even if the model equations are the same or very similar. The authors do not present a discussion that pins down the differences adequately. Another thought occurs: how do we know which of the solutions is the "truest" without an independent measure, e.g. from a natural dataset or a pointwise analytical solution or something comparable? The slope area plots imply that the topographies are equivalent in their properties. So what do the observed differences even mean?

*L.334:* Just an innocent comment, but I do hope that any decent model developer runs these test routinely before releasing a model and that it is not necessary to officially confirm this. Does not hurt to run these test especially when new to the software, but if everything is in order the main benefit was getting used to correctly handling the software.

*L.335:* Model accuracy strongly depends on grid and time spacing (e.g. the smaller, the more accurate), so most models should technically accomplish an arbitrary level of accuracy (in the bounds of round off and truncation errors). The question is not so much if a model can achieve a certain accuracy, but rather how long it takes to compute. It would have been nice if this aspect of the models was discussed.

*Section: variable time step simulations* I do not see the point in this demonstration with different timesteps. As mentioned before it is a fundamental property of discretization schemes that they are more accurate the finer the resolution, so differences in model results are expected. If the authors could present a method that allows to quantify the absolute accuracy of each model, the models could at least be ranked by the dt that is necessary to achieve a certain level of accuracy (computing time!).

*L.344:* I think this is the one part of the manuscript that troubles me the most. Although it is clearly stated that the solution is unstable, the results are repeatedly treated as if it is merely a matter of accuracy in the following part of the manuscript. The main difference is that in a stable simulation, the numerical approximation converges to the analytical solution and accuracy can be estimated and is well-behaved. Errors in unstable simulations accumulate over time and the results are essentially broken. The hole in the DEM shown in Fig. 5d is a very clear indicator (looks like NaN values?) and it should be clear that this data cannot be used, regardless of whether a part of the topography generated at smaller catchments sizes looks halfway plausible. Again, I also do not see what this demonstration adds to the topic of the paper. The shown stability issue is well-known for the type of model equation used here and the problem can be avoided by simply obeying the stability criterion.

*L. 358* I do not see what the authors mean by instability is "not an artefact of a voronoi grid". Is this meant to mean that the voronoi grid does not show the problem in an obvious way or that the instability is not a result from using a voronoi grid? The latter is not to be expected. The stability actually surprises me. Voronoi grids have a variable dx, so a slight shift in the stability criterion is to be expected depending on the combination of catchment area and dx, but this does not explain why the result looks so good. I assume that dt is lowered internally here as well to ensure stability, in which case it is not surprising at all that the Voronoi model works relatively well even if a much too large dt is used as input.

*L. 358* It is a hallmark of the implicit scheme that it has much better stability than the explicit scheme. The stability criterion is derived for explicit schemes, so it is not surprising that it is not

binding for an implicit scheme, either.

*L. 363* I assume that this issue arises mainly when starting from random values. Erosion rates are essentially random in the beginning. So the relative lowering of neighboring sites in one erosion step strongly depends on dt. Accordingly, the site that ends up as a lower neighbor at each timestep also depends on dt and how flow directions change then also depends on dt. I expect that this will have a discernible effect on divide migration and river network organisation. But what is the harm of that? If we generate topography from random values, the main goal is probably to obtain a valid steady state topography, not a specific network configuration.

*L. 377* Instability should not be mixed up with inaccuracy.

*L. 380* The smoothing of knickpoints is caused by numerical diffusion. I do not see a "sharp transition" even for lower timesteps and I do not expect to, since the problem cannot be solved by decrease in dt. That is the reason for the more complx knickpoint preserving algorithm that was also tested by the authors. Numerical diffusion is mentioned in the model decription there, but it should be brought into the right context again here.

*L. 390* Maybe the statement falsely connects to the next sentences, but: Since you establish in the following that the stair stepping is the result of 12.5 / 50 m instantaneous uplift at the beginning of each timestep, why is it a signal for numerical instability? Looks to me that the model does exactly what it is programmed to do. I assume that uplift for the whole timestep is added first, creating the large step. The subsequent part of the computation then calculates fluvial erosion in multiple steps for the same timestep (decreased dt for a stable solution). It looks to me that the knickpoint generated by the uplift is advected in upstream direction at roughly the correct velocity and it also falls closely to the predicted slope area relationship. What else can the model do but produce steps under these circumstances? I recall that stairstepping is actually a sign of reaching the threshold of stability, as it is also mentioned later in the manuscript (L. 469), but I do not think that this is the case here. Otherwise the authors should be clearer as to the nature of the signs of instability that they see in the diagram.

*L. 399* After the whole effort of using the stability criterion, it seems problematic if the models have the capability of reducing dt internally in order to intercept the error of the user and improve stability. It explains why so many of the results for too large dt look indecently good, but it also means that the dt put into the model is not the dt that was actually used and the observations regarding relative stability are tied to essentially unknown dt. Again it seems that the crucial algorithm that affects result is not properly discussed and interpreted.

*L. 443* The graphs for the lower dt that result in stable simulations are either overlapping or very close together, suggesting that the time to steady state does not vary significantly if dt is chosen within reasonable bounds. That the larger dt take longer to reach steady state is probably owed to the stairsteps that are carved into the landscape. That should take some additional time to smooth out. Since we already found several good reasons why these larger dt are very unpractical anyways, they should probably be left out of the comparison. The comparison is not on equal grounds and unjustifiably and artificially distorts the differences between the models.

*L. 461* I see similar oscillations in my own steady state topographies that are probably due to the same cause. I use a different software, but the same model equations and similar algorithms. The effect becomes more pronounced on larger grids. I do not think that the issue is a mark of inaccuracy, I rather suspect that the problem lies with the discretization of the grid. In a steepness index map, there are very localized ks anomalies of a few pixels size (both too high and

too low in direct neighborhood) at some of the peaks. It looks to me that the tributaries meeting at the peaks cannot achieve a steady state elevation that satisfies the steady state requirement for all streams simultaneously. I see a relatively fixed and small number of changes in flow directions and it seems to me that the grid just switches between two or maybe more states that oscillate around the unattainable ideal steady state, but the changes seem to be restricted to these tiny anomalous areas. I assume that it is mainly a consequence of a fixed gridspacing. Stream lengths cannot add up properly at the one end and the "residuals" concentrate at the drainage divide. A variable dx (or a finer subgrid that allows a better control for the location of the drainage divide) would probably solve the problem.

*L. 463* Time to steady state metrics are better comparable if dt is chosen in such a way that the accuracy of the numerical scheme is good enough that it does not significantly influence the results. I am not sure how sensible it is to do so as a concept as long as we do not have an independent measure what the "correct" timespan should be and while the reasons for delays are left largely undiscussed.

*L. 469* At this point it should be mentioned that accuracy comes at different levels and there is no black and white concept of wrong or right here. The smaller dt, the more accurate the solution (within the limits of round off and truncation errors, etc.). Also, this is one of several examples where I am unsure if the authors use a term in its mathematical/computational or colloquial sense.

*L. 474* Just no. Accuracy can be tweaked according to what we need as long as we know that the solution is well-behaved and follows the analytical solution, but instability is an avoidable and unpredictable error source and should always be reduced as far as possible.

*L. 495* As far as I see it the diagrams show quite clearly that numerical results obtained using a dt below the Courant criterion are very similar compared to results obtained with a larger dt... And just to clarify, the criterion only ensures stability of the numerical model and does not ensure a specific accuracy. The differences observed in the results for different dt can be eliminated How much farther dt must be reduced for a certain level of accuracy depends on the numerical scheme. I expect that the explicit scheme needs a higher reduction than shown here, but the results for the TRI, TRT, LVI and LHI overlap quite nicely and indicate that the model result probably would not change significantly anymore if dt was decrease.

---

## Author Response (AR1)

**Introduction and Summary**

We thank the two reviewers - Dr. Wickert and an anonymous reviewer (R1) - and two community members - Dr. Armitage and Dr. Madoff - for their thoughtful comments on our paper. We are encouraged that community members found the topic intriguing enough to participate in the review process without being asked. This suggests general interest for the core ideas we want to share with our community.

Despite this interest, it seems our paper in its current form did not clearly communicate our ideas as we intended. We included topics that some reviewers thought were unnecessary while excluding topics that other reviewers thought were necessary. To extremely briefly summarize how we interpret each reviewer's comments:

R1 : Reject the paper for many reasons including that the paper covers too many topics but not the ones that this reviewer thinks are most important. R1 also argues that what we suggest should become community practice is already LEM community practice (as discussed below, we disagree with R1's assessment of what is already LEM community practice). R1 suggests that we present mispractices that are not currently made in the LEM community.

Dr. Wickert : The paper is too long and yet doesn't give enough explanation of some of the surprising results. Similarly to R1, Dr. Wickert argues that some of what we say should be standard practice is already known by our community. However, Dr. Wickert likes the idea of a call to the community to develop benchmarks, just not our current form of it.

Dr. Armitage : There are many issues with LEMs that need to be discussed. If he were to prioritize issues with LEMs for benchmarking, Dr. Armitage might focus on a different set of issues than we highlighted (platform-specific numerical errors, grid resolution effects). However, Dr. Armitage seems to generally like the idea of a call to the community for clear benchmarks and user protocols.

Dr. Madoff : Although Dr. Madoff focuses on an even different set of issues than Dr. Armitage or we focused on, Dr. Madoff also seems to support an open discussion on LEM benchmarking. Dr. Madoff went even further to such other topics beyond benchmarking that the LEM community should address.

We find the fact that all the reviewers, in one way or another, liked the idea of community standards to indicate that this is a topic that needs to be discussed. However, our presentation and example choices did not impress any of the reviewers. As described below, our plan to deal with these revisions will effectively remove most of the components that both the formal reviewers and community commenters took issue with, and such, a formal rebuttal of the main points summarized above is not really warranted. Instead, we describe below our plan for a revised submission.

**Moving forward on this submission**

We accept that we may have tried to do both too much and not enough with this manuscript. Ultimately we would like to write a paper that will compel our community to agree on benchmarks and best practices for developing and using LEMs. However, in trying to motivate that, we have not presented what any of our reviewers think is a motivating case. In some ways that was part of our point - we (the authors) should not decide how to benchmark but the community should. However, we were not successful at getting our point across.

We would like to revise this submission by cutting out most of the manuscript and presenting only the results on time to steady state. This would allow us to resubmit something that better fits the description of a "short communication." It would also allow us to highlight these results, which we think are extremely important but were likely overlooked in our original submission. Based on discussions with community members outside of this review process, we think that the time to steady state results will be interesting and useful for many who use LEMs. Focusing on these results would allow us to more fully discuss why these results leave some of our previous assumptions in LEM studies on shaky ground. This would also open a discussion on the scope of what LEMs can and cannot do.

Notably, the time-to-steady-state comparison is a concrete example that motivates the need for the type of LEM benchmarking and intercomparison we hoped to motivate with our initial contribution. By focusing the revised contribution on this portion of our initial paper, we expect we will be able to document one example of why a community effort around benchmarking would be valuable.

Our general sense from R1 is that "mistakes" like we illustrated are not generally made (e.g., use of timesteps that are longer than stable under a courant condition). We disagree with this, but at the same time, we do not want, or feel it would be productive, to write a paper that catalogs the mistakes made by others in published work as motivation. However, to effectively rebut one of the primary criticisms of R1 would essentially require us to do such a cataloging. By writing a short contribution that is more focused on one specific issue, i.e., the variability in the time to steady-state in our experiments, we believe that we can use this as an illustrative example of the types of mistakes that can be made, explain how such mistakes fit in the context of the literature (without criticizing previous work), and use our own "mistake" as motivation for a commentary and call to the community on LEM benchmarking and best use. This new strategy will also allow us to more fully explore the dynamics of the variability in the time to steady-state observed in our experiments. As part of this revision, we would also then remove much of the content that R1 especially found superfluous, i.e., reviewing and definition of terms, etc.

Finally, we would like to thank the editors - Dr. Wolfgang Schwanghart and Dr. Andreas Lang - for helping us navigate the ESURF submission and review process. We recognize their sustained voluntary contributions to ESURF and our community.

---

## Referee Report (RR1)

**Peer-Review of**
**"Short Communication: The trouble with time to steady state calculated from computational landscape evolution models"**

Anonymous

January 24, 2024

**1 Summary of the Content**

The authors present a study that investigates the quality of time to steady state evaluation in numerical simulations of landscape evolution based on a detchment-limited erosion law.

**2 Overall Feedback**

The structure of the manuscript is easy to follow. However, information is not always presented in a clear and straightforward way. The authors seem to try to include all possible different viewpoints and to show e.g. the widest spectrum of possibilities of application for some methods (e.g. L. 33– 47). The authors also focus a great deal on the process of how they arrived at certain details of their model set-up instead of simply stating their choice and reason (e.g. L. 141–151). Considering that the manuscript is intended as short communication, there still is potential to shorten very detailed explanations to the quintessence of what is relevant to the topic.

Concerning the choice and "calibration" of methods, the connection between the chosen metrics and thresholds used for the determination of the time to steady state in numerical simulations versus the analytical assumption does not become clear. The authors focus on an analytical topographic steady state definition that assumes steady state is achieved when knickpoints fully migrated through the system. It is not discussed how knickpoint migration affects their metrics and how this can be used as alternative method to refine their measurement of numerical time to steady state $T_E$. The question of acceptable variability in the chosen metrics is raised, but not discussed with regards to the magnitude of fluctuations due to numerical artefacts or ongoing small-scale rearrangements of flow directions that complicate the automatic detection of response time in numerical simulations.

In that regard, I think the usage of the terms "response time" and "steady state" should be used non-synonymously where the numerical model is concerned. The analytical solution $T_A$ for steady state is defined to be equal to the response time, but I think the central result that can be drawn from the evaluation of the numerical simulations is that the same does not apply there. I think the logical conclusion is that two times need to be derived from the metrics, one for response time (also based on assummptions on knickpoint migration) and one for steady state in its stricter sense ($\frac{\Delta h}{\Delta t} = 0$).

I do not agree with the final conclusion that numerical simulations are basically unsuited to derive response times. It is certainly true that the most simple method of automatic detection does not work, but that problem can be approached by separating between model behavior and numerical artefacts and finding a refined method for measurement that is less sensitiv to the latter.

The other question is how much we need to depend on numerical simulations to determine response time under detachment-limited conditions, when an easy-to-use analytical solution does exist for that case. I appreciate the effort of the authors to investigate the applicability of landform evolution models (LEMs), but in many parts of the manuscript the question arises how much more important it would be to include different erosion laws into the comparison.

I think the manuscript needs at least a serious overhaul regarding the evaluation and discussion if it should be considered for publication. The findings of the authors differ significantly from published literature regarding the reliability of the investigated models, but the causes are not identified or discussed properly. I assume that most of the issues disappear if the evaluation of the metrics is revised and that the results will mostly reflect what is already published on the behavior of these models. Based on the introduction, I initially expected that the authors at least provide a recommendation how to deal with numerical artefacts, but I was disappointed that the evaluation is based on what I perceive to be a mostly arbitrary threshold that has nothing to do with the chosen analytical estimation for response time. I really regret to recommend rejection.

**3 General Remarks**

- I still miss the discussion of the applicability of detachment-limited erosion (and the derived theoretical steady state criterion presented in Eq. 9 and 10) where the interpretation of natural landscapes is concerned. I raise this point (again) because the evaluation of natural landscapes is explicitly stated as main motivation in the introduction and because the authors insist on a very low and strict threshold for measuring time to steady state in their numerical models. My main points are these:

    - Sediment transport and other secondary erosion processes (hillslope processes, aeolian processes, etc.) have a significant influence on response times (i.e. knickpoint velocity). The presented steady state criterion is based on the theoretical time a knickpoint needs to travel through the entire catchment (according to Whipple (2001)). Doesn't this criterion only hold for purely detachment-limited conditions?

    - Is it not to be expected that variations due to different erosion laws are much higher in magnitude than the variations shown here between different numerical implementations of the same erosion law investigated here? What is your estimation to that regard? I expect that the correct interpretation of natural landscapes depends much more on the choice of the correct erosion law than on differences purely due to numerical implementation.

    - The argument that natural landscapes might behave different from what the model assumes is raised in the end (L. 307f), seemingly to deflect from the very high apparent discrepancy between theoretical response time and response/steady state time measured from the simulations. But the problem seems to lie in the method of evaluation and not model choice there, the analytical solution used for comparison is essentially derived from the basic model equation (with some integration and addition of Hack's empirical observation) and should be reproducable with the numerical model. The topic of initial model choice should be discussed right at the start.

  I really only encourage to add a short clarification of the limitations where the importance of results of this study are concerned, not a detailed overview. Some assumptions for the evaluation of steady state/response times presented here probably cannot be transferred directly to other LEMs using different erosion laws.

- In response to the reaction to my first review: I certainly did not wish to imply that errors in implementation and interpretation are not made by the user community. The point I tried to make is that the mathematical theory to correctly predict e.g. accuracy and stability of numerical schemes, i.e. the explicit scheme, already exists for several decades and it would have been very useful to apply this theory in the context presented in the last version of the manuscript, especially when a younger generation of model users without a rigorous mathematical background is addressed. It is my suspicion that model developers, especially when coming from a strong mathematical background, consider such knowledge basic on a textbook level and omit the details in model descriptions "because it is self-evident". The scope of the manuscript has changed, so I expect no formal answer, just felt the need to clarify why I am very insistent on some of what I consider to be fundamental points.

**4 Major Remarks**

**4.1 Introduction**

- **L. 34–47** It seems your study only considers a one-time change in conditions and does not contribute to the scenarioes where conditions change continuously or faster than the response time. Maybe this part can be shortened accordingly?

**4.2 Section 4 Experimental set-up**

- **L. 107–110** If you know that the landscape was static, it seems that you formally measured something. Maybe something along the line "Initial conditions were static according to the criterion established in the discussion" is sufficient here?

- **L. 116f** Maybe explain the general setup and similarities between models before mentioning specific exceptions to simplify comprehension (from general to detail, so to say).

- **L. 125** Is that simply a circumscription for Delauney triangulation? What is the average variation and what do you mean with "variation in a regular way" if the grid is actually irregular?

- **L. 135** correct within numerical errors, absolute accuracy with regard to the analytical solution usually still depends on dt (not only on machine precision/rounding errors).

- **L. 141–149** The quintessence seems to be that the largest possible catchment area was deliberately overestimated to ensure a stable simulation, and rounded to have a nice dt value. This passage can be shortened.

**4.3 5.1 Numerically modeled**

- **L. 196** Looks like Equation (8) is essentially the temporal derivative of equation (6)? Considering that Figures 1 and 3 are hard to decipher because of the many subplots and that the graphs of both metrics seem to deliver the same information regarding time to steady state, one of them could maybe be discarded to gain more space?

- **L. 210** I think the graph should be interpreted by clearly dividing between the causes that are affecting the behaviour (e.g. what is the actual model behaviour and what are numerical artefacts). The first part of the graph (up to roughly $T_A$) seems clearly affected by knickpoint migration and reflects where most of the adjustment of erosion rates towards a new steady state should be happening, and the second part seems to be dominated by numerical artefacts and some effects of (potentially related?) divide migration as suggested by the authors (possibly masking some knickpoint-related behaviour?).

  Concerning the first, Whipple and Tucker (1999) and Whipple (2001) state that the equation for theoretical time to steady state is derived from the response time of catchments that is controlled by knickpoint migration (speed at which information can be transmitted through the system), and that steady state can at the earliest be achieved when knickpoints have migrated through the entire system. I propose that the metrics are quite clearly affected by knickpoint migration and that the change in behaviour of the graphs from relatively smooth trends to more chaotic swings occurs when knickpoints arrive at the divide in your modeled scenarios.

  I assume that the second phase is mainly influenced by what I will refer to as "numerical artefacts" for simplicity, because I strongly suspect that discretization of the grid is the driving factor that keeps local rearrangement of flow directions going on for such a long time. I stated before that I often observe a localized switching of flow directions. The reason seems to be due to the discretization of the grid that does not allow exact adjustment of stream lengths to the ideal equilibrium length (since increments are limited by the gridspacing and there is also competition between catchments). At least anomalies in the steepness ks that appear as a group of too low and too high ks values seem to support that assumption. I would be interested if you observe the same in your simulations, or if you observe a different cause.

  For the evaluation, I also think it should be considered that the variations occurring after the analytical response time are several magnitudes lower than the differences in the initial phase of your graphs. They are measurable in a computer simulation, but are they significant enough to be seriously regarded or should we find a way to filter them out of our results, especially if they might be numerical aretefacts and compared to all the much larger uncertainties that probably arise from parameter estimation or choice of erosion law? In summary, I suggest separating between response time and time to steady state in numerical simulations and evaluate both times.

- **L. 214f** I cannot find a good explanation for the particular choice of these threshold values in the following. They remain arbitrary. Please state the criterion that you used. If you just used the lowest threshold that presented itself to you and used it for all simulations, how should a person choose a threshold if another software is used? Also, it seems that some metrics could use a higher threshold. This would partly significantly affect the results you presented in the following.

- **L. 215** "Conservative" seems an understatement. Less than 0.1 nanometer vertical change per year does not seem to be a practical threshold for steady state, especially not where comparison to natural landscapes is concerned.

- **L. 235** I understand that $k_a$ and $h$ are not supposed to change much by divide migration. But how large is the variation between different catchments across the grid and does this introduce an error for the time to steady state calculation (especially where the detection of knickpoint migration in the simulations is concerned)? Did you perchance test the evaluation of (maybe only a few of the largest) individual catchments with the respective individual geometric parameters to test if this improves the detectability of response time/steady state? On that note, I believe that Whipple observed that vertical knickpoint velocity equals uplift rate, so could that not be used for a response time estimation that is less dependent on additional parameters that potentially add uncertainties to the evaluation?

**4.4 6 Time to steady state results**

- **L. 250** Is this a counterargument for using the knickpoint criterion? A systematic deviation in dependence of timestep is theoretically to be expected. Depending on the numerical scheme rates of change are systematically over or underestimated and the effect becomes usually worse with larger timesteps.

- **L. 251f** Not clear if this is intended as a counterargument, or why you expect a smooth decrease for maximum elevation. The change in maximum elevation seems to behave as it theoretically should. Erosion rates are adjusted only below the knickpoint, mainly by adjustment of the slopes. Above the knickpoint all nodes including the highest peak are constantly eroded with the old erosion rate and at the same time elevated with the new uplift rate. Relative topography should not be affected by baselevel drop/baselevel increase until the knickpoint passes (so the maximum elevation can be expected to remain perfectly stationary at first). The resulting constant net elevation increase (40 m/1e5yr) of the peak seems to be reflected in your graph. A sudden change/drop in your metric is to be expected at the time when knickpoints arrive at the peak, which should also include the last knickpoint in the system since the peak should also be associated with the longest stream and longest knickpoint travel time.

- **Figure 1** is not readable at all. Beside reduction of the number of subplots, I also suggest that a magnification of the initial phase (until the analytical time to steady state) is provided. In the current state it is not possible to judge properly how the behaviour of the graphs changes at this important time because the interesting part is squished to the left boundary. Have you tried a double-logarithmic plot? There are several solid black lines in some of the diagrams where there should only be one according to the figure decription. Also, you include an additional analytical time to steady state based on an average river length, but do not explain or discuss it in the main text. I would be very interested how you calculated this, especially because this estimation of response time is a perfect fit to the empirical response time in Fig. 3 for max elevation and the knickpoint preserving algorithm (where I expect that empirical response time should be closest to $T_A$ based on the specifics of numerical implementation).

- **L. 261** Can you include an explanation why the Voronoi grid suffers less from network rearrangement? The advantages/disadvantages of the Voronoi are not discussed before, although it seems from the model descriptions that it was deemed very important to include it. I recall that Voronoi grids are better suited for the representation of natural landscapes because they are more flexible, but need more effort to generate because triangles near the boundary tend to be disproportionly long.

- **L. 265** $T_E$ as measured by the threshold does probably not represent response time in the sense of $T_A$. Also, would it not be better to just evaluate the largest catchment (that was also used to to estimate $T_A$) if is to be expected that variation of $k_a$, $h$ and $L$ add uncertainty to the results? Every metric except maximum elevation is probably affected because they would essentially middle between all the different knickpoint arrivel times.

- **L. 270** How are the numerical methods on a Voronoi grid different if they are supposed to be an implicit and explicit method respectively?

- **L. 273** If a threshold does not work reliably, have you considered another approach better suited to detect response times? It seems to me that using the temporal derivative of maximum elevation would work very good to find the time when knickpoints start to arrive at the peak.

- **L. 278** Do you assume that Braun and Willett (2013) are wrong or do you have a hypothesis why your results are so different? Even if you find that your results depend on initial flow directions, you seem not to be able to reproduce the relationship that you found in the literature?

- **Figure 2** Please include the results of the time of completed knickpoint migration as occurring in the simulations versus analytical response time (Eq. 9).

- **298** Your results seem to nicely show the effect of numerical diffusion/knickpoint smearing as well. TRT reduces knickpoint smearing and shows the nicest result for maximum elevation.

- **Figure 3** I am puzzled by what appears to be a solid double line in the first row. There also is a sentence fragment at the end of the figure description. Also, your derived results here appear to be particularly sensitiv to your choice of threshold, and I do not see why a higher threshold should not be used.

- **L. 303** I was under the impression that your main conclusion is that the metric combined with your single threshold is not suitable at all to detect response time.

- **L. 311**
  - Don't you state here that the assumption of detachment-limited erosion is inaccurate where the interpretation of natural landscapes is concerned, thereby devaluating your choice of numerical model (at least as far as the motivation for this study as stated in the introduction is concerned)?

  - I think the more important question in this context is how to formulate the right criterion to reliably measure response times in this type of numerical model. Yes, the estimation for steady state presented in Eq. 9 might not be the ideally representative criterion where natural landscapes are concerned,

but the numerical model is still required to reproduce predictions that are essentially derived from its model equations (if not, it is either an error in implementation or a numerical artefact). It seems quite clear that some unexpected fluctuations prevent perfect steady state ($\frac{\Delta h}{\Delta t} = 0$ during simulations here, so model topographic steady state in the strictest sense obviously does not equal response time as predicted by knickpoint migration. I am surprised that you do not discuss refinement of the evaluation, e.g. by considering the potential effect of numerical artefacts and thresholding accordingly, trying to combine metrics or by changing metrics to make them better suited to the detection of knickpoints, maybe try to focus evaluation on single catchments to achieve a better precision.

- **L. 327** At this point I think it becomes essential to discuss the possibility of numerical artefacts on the results of this study. It is tempting to see the counterpart of a natural process, but the cause of numerical artefacts are usually unintentionally an caused purely by limitations of numerical implementation and cannot be transferred to nature (nature is not limited by a discrete grid spacing, for example)

- **L. 334** I think it is explicitly stated that $T_A$ is to be onsidered a minimum estimate in Whipple (2001). The influence of divide migration and resulting error on $T_A$ can be estimated (or monitored during your simulations). Do you have any indication that the large discrepancy you present can realistically be explained by the effect of divide migration or network instability, as proposed in L. 316?

**5 Minor Remarks**

- **L. 63** "voronoi" is a proper noun I think?

- **Table 1** Maybe change to "implicit (Fastscape)" in the last column for uniform style.

- **L. 137** "$v$ is the speed *at which*"?

- **L. 167** Waht do you mean by "empirical model"? Not the numerical model?

- **Eq.5** $i$ is the index of the node, I presume?

---

## Author Response (AR2)

We are grateful for the time that volunteers have put in to review our paper. Reviewers have been thoughtful and thorough. Our paper is now stronger thanks to their comments. Our responses to their comments are shown in red below. In cases when the comments were long, we bolded what we thought was the essence of the comment for us to respond to.

Sincerely,
Nicole Gasparini, for all authors.

**Reviewer 1 comments:**

2 Overall Feedback

The structure of the manuscript is easy to follow. However, information is not always presented in a clear and straightforward way. The authors seem to try to include all possible different viewpoints and to show e.g. the widest spectrum of possibilities of application for some methods (e.g. L. 33– 47). The authors also focus a great deal on the process of how they arrived at certain details of their model set-up instead of simply stating their choice and reason (e.g. L. 141–151). **Considering that the manuscript is intended as short communication, there still is potential to shorten very detailed explanations to the quintessence of what is relevant to the topic.**

We find that most modelling studies do not provide enough information to recreate the experiments. We prefer to explain what we did and let reviewers decide to skip it than not make information available, so that reviewers cannot recreate what we did or understand why we did it.

Concerning the choice and "calibration" of methods, the **connection between the chosen metrics and thresholds used for the determination of the time to steady state in numerical simulations versus the analytical assumption does not become clear.**

We now have a motivation section that states how other papers describe methods used to reach steady state. We hope this will make it more clear why we have chosen the metrics we have chosen. The short answer is that essentially there are no established metrics.

**The authors focus on an analytical topographic steady state definition that assumes steady state is achieved when knickpoints fully migrated through the system**. It is not discussed how knickpoint migration affects their metrics and how this can be used as alternative method to refine their measurement of numerical time to steady state TE. The question of acceptable variability in the chosen metrics is raised, but not discussed with regards to the magnitude of fluctuations due to numerical artefacts or ongoing small-scale rearrangements of flow directions that complicate the automatic detection of response time in numerical simulations.
In that regard, **I think the usage of the terms "response time" and "steady state" should be used non-synonymously where the numerical model is concerned. The analytical solution TA for steady state is defined to be equal to the response time, but I think the central result that can be drawn from the evaluation of the numerical simulations is that the same does not apply there.** I think the logical conclusion is that two times need to be derived from the metrics, one for response time (also based on assummptions on knickpoint migration) and one for steady state in its stricter sense ( h t = 0).

We have added more discussion of the difference between topographic steady state and time to steady state, as predicted by the analytical equation.

I do not agree with the final conclusion that numerical simulations are basically unsuited to derive response

times. It is certainly true that the most simple method of automatic detection does not work, but **that problem can be approached by separating between model behavior and numerical artefacts and finding a refined method**
**for measurement that is less sensitiv to the latter.**

We are encouraged that the reviewer thinks the problem is tractable. We hope that the community will recognize the "nuances" of model behavior in the future, which is why we wrote this paper. We have not seen such a refined method as the reviewer describes used before, and we would appreciate a reference for the publication that describes such methods.

**The other question is how much we need to depend on numerical simulations to determine response time under**
**detachment-limited conditions, when an easy-to-use analytical solution does exist for that case**. I appreciate the effort of the authors to investigate the applicability of landform evolution models (LEMs), but in many parts of the manuscript the question arises how much more important it would be to include different erosion laws into the comparison.

We do not argue against the use of the analytical solution. We only mean to illustrate that one needs to be aware that determining steady state, and time to steady state using a model, should be done with caution. Because many model results are presented on a "steady state" landscape, it is important for the community to reach agreement as to what this means and not over interpret numerical artifacts.

Using different erosion laws is beyond the scope of this study.

I think the manuscript needs at least a serious overhaul regarding the evaluation and discussion if it should be considered for publication. **The findings of the authors differ significantly from published literature regarding**
**the reliability of the investigated models, but the causes are not identified or discussed properly**. I assume that most of the issues disappear if the evaluation of the metrics is revised and that **the results will mostly reflect what is already published on the behavior of these models**. Based on the introduction, I initially expected that the authors at least provide a recommendation how to deal with numerical artefacts, but **I was disappointed that the evaluation is based on what I perceive to be a mostly arbitrary threshold that has nothing to do with the chosen**
**analytical estimation for response time**. I really regret to recommend rejection.

We are not aware of and could not find the published literature regarding the reliability of our study models. We wish the reviewer would have provided references. We are unsure what the reviewer means when they say "the results will mostly reflect what is already published on the behavior of these models". Without references, it is difficult for us to respond to this comment.

We removed the arbitrary threshold.

3 General Remarks
• I still miss the discussion of the applicability of detachment-limited erosion (and the derived theoretical steady state criterion presented in Eq. 9 and 10) where the interpretation of natural landscapes is concerned.
**I raise this point (again) because the evaluation of natural landscapes is explicitly stated as main motivation in the introduction and because the authors insist on a very low and strict threshold for measuring time to steady state in their numerical models.** My main points are these:

Many studies use detachment limited models to understand natural landscapes. We have removed our "very low and strict threshold".

– Sediment transport and other secondary erosion processes (hillslope processes, aeolian processes, etc.) have a significant influence on response times (i.e. knickpoint velocity). The presented steady state criterion is based on the theoretical time a knickpoint needs to travel through the entire catchment (according to Whipple (2001)). **Doesn't this criterion only hold for purely detachment-limited conditions?**

We are unsure what the reviewer is saying. We use a detachment-limited model, and we compare it with the analytical solution for the detachment limited model. We have made changes that clarify that knickpoint propagation time and modeled time to steady state are not necessarily the same thing.

**– Is it not to be expected that variations due to different erosion laws are much higher in magnitude than the variations shown here between different numerical implementations of the same erosion law investigated here?** What is your estimation to that regard? I expect that the correct interpretation of natural landscapes depends much more on the choice of the correct erosion law than on differences purely due to numerical implementation.

Possibly variations from different erosion laws would be higher in magnitude. In our mind, that makes it even more compelling that the community come to agreement about what they are measuring.

– The argument that natural landscapes might behave different from what the model assumes is raised in the end **(L. 307f),** seemingly to deflect from the very high apparent discrepancy between theoretical response time and response/steady state time measured from the simulations. But the problem seems to lie in the method of evaluation and not model choice there, the analytical solution used for comparison is essentially derived from the basic model equation (with some integration and addition of Hack's empirical observation) and should be reproducable with the numerical model. The topic of initial model choice should be discussed right at the start.

The paragraph referred to above was deleted. We have added more discussion about the differences between the analytical solution and the model behavior.

**I really only encourage to add a short clarification of the limitations where the importance of results of this study are concerned, not a detailed overview. Some assumptions for the evaluation of steady state/response times presented here probably cannot be transferred directly to other LEMs using different erosion laws.**

We are not quite sure what the reviewer is asking for here. This may or may not address this comment, but we did add a Motivation section to the paper that discusses the lack of agreement or standard processes within the community. (At least there is a lack of agreed process in published studies that we are aware of.)

• In response to the reaction to my first review: I certainly did not wish to imply that errors in implementation and interpretation are not made by the user community. The point I tried to make is that the mathematical theory to correctly predict e.g. accuracy and stability of numerical schemes, i.e. the explicit scheme, already exists for several decades and it would have been very useful to apply this theory in the context presented in the last version of the manuscript, especially when a younger generation of model users without a rigorous mathematical background is addressed. It is my suspicion that model developers, especially when coming from a strong mathematical background, consider such knowledge basic on a textbook level and omit the details in model descriptions "because it is self-evident". The scope of the manuscript has changed, so **I expect no formal answer**, just felt the need to clarify why I am very insistent on some of what I consider to be fundamental points.

No answer is provided.

4 Major Remarks
4.1 Introduction
• L. 34–47 It seems your study only considers a one-time change in conditions and does not contribute to the scenarioes where conditions change continuously or faster than the response time. Maybe this part can be shortened accordingly?

We are not sure what the reviewer is asking of us.

4.2 Section 4 Experimental set-up
• L. 107–110 If you know that the landscape was static, it seems that you formally measured something.

Maybe something along the line "Initial conditions were static according to the criterion established in the discussion" is sufficient here?

We did not change anything.

• L. 116f Maybe explain the general setup and similarities between models before mentioning specific exceptions to simplify comprehension (from general to detail, so to say).

We did not change anything.

• L. 125 Is that simply a circumscription for Delauney triangulation? What is the average variation and what do you mean with "variation in a regular way" if the grid is actually irregular?

Changed to : The numerical experiments that use a Voronoi grid are created from a Delaunay triangulation of staggered rows of nodes spaced 100 m apart in the x direction.

• L. 135 correct within numerical errors, absolute accuracy with regard to the analytical solution usually still depends on dt (not only on machine precision/rounding errors).

Yes. We did not change anything.

• L. 141–149 The quintessence seems to be that the largest possible catchment area was deliberately overestimated to ensure a stable simulation, and rounded to have a nice dt value. This passage can be shortened.

Shortened.

4.3 5.1 Numerically modeled
• L. 196 Looks like Equation (8) is essentially the temporal derivative of equation (6)? Considering that Figures 1 and 3 are hard to decipher because of the many subplots and that the graphs of both metrics seem to deliver the same information regarding time to steady state, one of them could maybe be discarded to gain more space?

We have changed the figures to include two that are easier to read. We kept versions of Figures 1 and 3 for completion.

• L. 210 I think the graph should be interpreted by clearly dividing between the causes that are affecting the behaviour (e.g. what is the actual model behaviour and what are numerical artefacts). The first part of the graph (up to roughly TA) seems clearly affected by knickpoint migration and reflects where most of the adjustment of erosion rates towards a new steady state should be happening, and the second part seems to be dominated by numerical artefacts and some effects of (potentially related?) divide migration as suggested by the authors (possibly masking some knickpoint-related behaviour?).

We are unsure what the reviewer is asking of us.

Concerning the first, Whipple and Tucker (1999) and Whipple (2001) state that the equation for theoretical time to steady state is derived from the response time of catchments that is controlled by knickpoint migration (speed at which information can be transmitted through the system), and that steady state can at the earliest be achieved when knickpoints have migrated through the entire system. I propose that the metrics are quite clearly affected by knickpoint migration and that the change in behaviour of the graphs from relatively smooth trends to more chaotic swings occurs when knickpoints arrive at the divide in your modeled scenarios. I assume that the second phase is mainly influenced by what I will refer to as "numerical artefacts" for simplicity, because I strongly suspect that discretization of the grid is the driving factor that keeps local rearrangement of flow directions going on for such a long time. I stated before that I often observe a localized switching of flow directions. The reason seems to be due to the discretization of the grid that does not allow exact adjustment of stream lengths to the ideal equilibrium length (since increments

are limited by the gridspacing and there is also competition between catchments). At least anomalies in the steepness ks that appear as a group of too low and too high ks values seem to support that assumption. I would be interested if you observe the same in your simulations, or if you observe a different cause. For the evaluation, I also think it should be considered that the variations occurring after the analytical response time are several magnitudes lower than the differences in the initial phase of your graphs. They are measurable in a computer simulation, but are they significant enough to be seriously regarded or should we find a way to filter them out of our results, especially if they might be numerical aretefacts and compared to all the much larger uncertainties that probably arise from parameter estimation or choice of erosion law? In summary, I suggest separating between response time and time to steady state in numerical simulations and evaluate both times.

We have removed the strict interpretation of experimental time to steady state.

• L. 214f I cannot find a good explanation for the particular choice of these threshold values in the following. They remain arbitrary. Please state the criterion that you used. If you just used the lowest threshold that presented itself to you and used it for all simulations, how should a person choose a threshold if another software is used? Also, it seems that some metrics could use a higher threshold. This would partly significantly affect the results you presented in the following.

We no longer use this threshold value.

• L. 215 "Conservative" seems an understatement. Less than 0.1 nanometer vertical change per year does not seem to be a practical threshold for steady state, especially not where comparison to natural landscapes is concerned.

We no longer use this threshold value.

• L. 235 I understand that ka and h are not supposed to change much by divide migration. But how large is the variation between different catchments across the grid and does this introduce an error for the time to steady state calculation (especially where the detection of knickpoint migration in the simulations is concerned)? Did you perchance test the evaluation of (maybe only a few of the largest) individual catchments with the respective individual geometric parameters to test if this improves the detectability of response time/steady state? On that note, I believe that Whipple observed that vertical knickpoint velocity equals uplift rate, so could that not be used for a response time estimation that is less dependent on additional parameters that potentially add uncertainties to the evaluation?

We did not do more testing of the hack parameters. We did not do further tests of estimates of response time.

4.4 6 Time to steady state results
• L. 250 Is this a counterargument for using the knickpoint criterion? A systematic deviation in dependence of timestep is theoretically to be expected. Depending on the numerical scheme rates of change are systematically over or underestimated and the effect becomes usually worse with larger timesteps.

This was deleted.

• L. 251f Not clear if this is intended as a counterargument, or why you expect a smooth decrease for maximum elevation. The change in maximum elevation seems to behave as it theoretically should. Erosion rates are adjusted only below the knickpoint, mainly by adjustment of the slopes. Above the knickpoint all nodes including the highest peak are constantly eroded with the old erosion rate and at the same time elevated with the new uplift rate. Relative topography should not be affected by baselevel drop/baselevel increase until the knickpoint passes (so the maximum elevation can be expected to remain perfectly stationary at first). The resulting constant net elevation increase (40 m/1e5yr) of the peak seems to be reflected in your graph. A sudden change/drop in your metric is to be expected at the time when knickpoints arrive at the peak, which should also include the last knickpoint in the system since the peak should also be associated with the longest stream and longest knickpoint travel time.

This was clarified.

• Figure 1 is not readable at all. Beside reduction of the number of subplots, **I also suggest that a magnification of the initial phase (until the analytical time to steady state) is provided.** In the current state it is not possible to judge properly how the behaviour of the graphs changes at this important time because the interesting part is squished to the left boundary. Have you tried a double-logarithmic plot? There are several solid black lines in some of the diagrams where there should only be one according to the figure decription. Also, you include an additional analytical time to steady state based on an average river length, but do not explain or discuss it in the main text. I would be very interested how you calculated this, especially because this estimation of response time is a perfect fit to the empirical response time in Fig. 3 for max elevation and the knickpoint preserving algorithm (where I expect that empirical response time should be closest to TA based on the specifics of numerical implementation).

We magnified the initial time of two sets of experiments and made these figures 1 and 2. We kept the former figure 1 (it is now figure 3) because it seemed inappropriate to not illustrate all of the results. I find log time plots difficult to interpret, so I opted not to make that change. We removed the lines from these plots.

• L. 261 Can you include an explanation why the Voronoi grid suffers less from network rearrangement? The advantages/disadvantages of the Voronoi are not discussed before, although it seems from the model descriptions that it was deemed very important to include it. I recall that Voronoi grids are better suited for the representation of natural landscapes because they are more flexible, but need more effort to generate because triangles near the boundary tend to be disproportionally long.

We did not include an explanation of why the Voronoi grid has less rearrangement. We are not 100% sure why. It is not "very important" to include the Voronoi results, but we advocate for better understanding for the impact of different grid types on numerical modeling. Most of that aspect of our study is removed from this version of the paper because the reviewers were not supportive of the first version of this paper. But we kept in the Voronoi results because they help illustrate the impact that network rearrangement has on time to steady state. We can still remove these results though, if this version of the paper is invited for further revisions.

• L. 265 TE as measured by the threshold does probably not represent response time in the sense of TA. Also, would it not be better to just evaluate the largest catchment (that was also used to to estimate TA) if is to be expected that variation of ka, h and L add uncertainty to the results? Every metric except maximum elevation is probably affected because they would essentially middle between all the different knickpoint arrival times.

We removed T_e calculations.

• L. 270 How are the numerical methods on a Voronoi grid different if they are supposed to be an implicit and explicit method respectively?

This was deleted, but good catch. We meant the flow routing algorithms are different.

• L. 273 If a threshold does not work reliably, have you considered another approach better suited to detect response times? It seems to me that using the temporal derivative of maximum elevation would work very good to find the time when knickpoints start to arrive at the peak.

We removed the threshold approach.

• L. 278 Do you assume that Braun and Willett (2013) are wrong or do you have a hypothesis why your results are so different? Even if you find that your results depend on initial flow directions, you seem not to be able to reproduce the relationship that you found in the literature?

No, of course we do not think that Braun and Willett are wrong!

• Figure 2 Please include the results of the time of completed knickpoint migration as ocurring in the

simulations versus analytical response time (Eq. 9).

We did not calculate the time of completed knickpoint migration.

• 298 Your results seem to nicely show the effect of numerical diffusion/knickpoint smearing as well. TRT reduces knickpoint smearing and shows the nicest result for maximum elevation.

We are unsure of what the reviewer is asking of us.

• Figure 3 I am puzzled by what appears to be a solid double line in the first row. There also is a sentence fragment at the end of the figure description. Also, your derived results here appear to be particularly sensitiv to your choice of threshold, and I do not see why a higher threshold should not be used.

Threshold was eliminated. Lines from figure 3 were deleted.

• L. 303 I was under the impression that your main conclusion is that the metric combined with your single threshold is not suitable at all to detect response time.

That text was changed.

• L. 311
– Don't you state here that the assumption of detachment-limited erosion is inaccurate where the interpretation of natural landscapes is concerned, thereby devaluating your choice of numerical model (at least as far as the motivation for this study as stated in the introduction is concerned)?

That text was deleted.

– I think the more important question in this context is how to formulate the right criterion to reliably measure response times in this type of numerical model. Yes, the estimation for steady state presented in Eq. 9 might not be the ideally representative criterion where natural landscapes are concerned, but the numerical model is still required to reproduce predictions that are essentially derived from its model equations (if not, it is either an error in implementation or a numerical artefact). It seems quite clear that some unexpected fluctuations prevent perfect steady state ( h
 t = 0 during simulations
here, so model topographic steady state in the strictest sense obviously does not equal response time as predicted by knickpoint migration. I am surprised that you do not discuss refinement of the evaluation, e.g. by considering the potential effect of numerical artefacts and thresholding accordingly, trying to combine metrics or by changing metrics to make them better suited to the detection of knickpoints, maybe try to focus evaluation on single catchments to achieve a better precision.

I think some of our changes are along these lines.

• L. 327 At this point I think it becomes essential to discuss the possibility of numerical artefacts on the results of this study. It is tempting to see the counterpart of a natural process, but the cause of numerical artefacts are usually unintentionally an caused purely by limitations of numerical implementation and cannot be transferred to nature (nature is not limited by a discrete grid spacing, for example)

We now mention numerical artifacts.

• L. 334 **I think it is explicitly stated that TA is to be onsidered a minimum estimate in Whipple (2001).**
The influence of divide migration and resulting error on TA can be estimated (or monitored during your simulations). Do you have any indication that the large discrepancy you present can realistically be explained by the effect of divide migration or network instability, as proposed in L. 316?

Yes, you are correct. We have clarified that.

5 Minor Remarks
• L. 63 "voronoi" is a proper noun I think?
Fixed.

• Table 1 Maybe change to "implicit (Fastscape)" in the last column for uniform style.

Done.

• L. 137 "v is the speed at which"?

Changed.

• L. 167 Waht do you mean by "empirical model"? Not the numerical model?

Fixed.

• Eq.5 i is the index of the node, I presume?

Fixed.

**Reviewer 2**

I am very interested in the finding that drainage rearrangement drives the longer-than-theoretical and inconsistent landscape response times. I comment on this a bit more below, but I have a couple of thoughts regarding possibliities. I am not sure if you will find them in-scope or out-of-scope, and leave this for the authors to decide.

First, **I thought of Kwang et al. (2021), who note that the lateral dynamics of rivers can be quite important in sustaining a dynamic, rather than static, equliibrium topography. Do you think that "noise" in erosion rates caused by this process could further delay the approach to equilibrium beyond the Whipple & Tucker (1999) analytical calculation?** This might further underscore your point towards the end of the article: even though the lateral-erosion component is not included in your tested LEMs, adding more expected realism might further separate from the theory rather than making the model closer. What do you think?
*Note that I was Jeffrey Kwang's postdoc advisor and a graduate-student colleague with his co-author Abby Langston. Please do not feel any pressure to cite this work; there may be other papers out there, and I just know this one.*

Yes, we do think these other processes would impact time to steady state. We have added this to our discussion.

Second, I considered how drianage basins form in the first place, as the Whipple & Tucker (1999) theoretical predction is based on the idea that the channel already exists. However, we see that the initial integration of drainage networks can produce lags in landscape evolution (cf. Lai & Anders, 2018: in this case, even when flow across internal depressions is allowed).

I then wondered about the more fundamental difference between propagation of a signal up an existing river and catchment (what Whipple & Tucker, 1999, were considering for "response time") and the time it takes for that order to co-develop alongside the signal propagation (what you are measuring in the models, albeit, with the additional consideration of what the models themselves are doing). This is all quite interesting, to me, in two specific ways. First, it lines up with some recent work demonstrating that cross-basin draingae integration may likely occur via overflow events (Hilgendorf et al., 2020) and that headward erosion is unlikely unless a channel already exists. Second, it underscores that we may have "noise" in how real landscapes respond to perturbations because of the nonlinear interactions of drainage sub-basins playing out atop whatever initial conditions existed, potentially giving both limit and geological guidance to geomorphic predictions.

We interpret this text as comments without anything for us to change or add.

[minor] The article is 342 lines of text+equations, with three large figures and one table. I checked the definition of "short communication" and confirmed my memory that this manuscript type should be "a few pages only". I would see it as a considerable overstep of my role here to discuss which kind of manuscript it *should* be, but I want to point this out to the editors and authors.
https://www.earth-surface-dynamics.net/about/manuscript_types.html

We leave it to the editors if this qualifies as a short communication or not.

LINE BY LINE

"The sensitivity of time to steady state to computational time step is not consistent among models or even within a single model."
The 2x "to" make the meaning ambiguous. I suggest:
"Time to steady state varies inconsistently with time-step length, both within a single model and among different models."
Is that right? I hope so. Otherwise I really didn't understand.

This sentence was changed as suggested.

95 (and other equations)
[minor] I think that ESurf has commas or other punctuation after equations, such that they fit into the surrounding sentence structure

I will wait for the copy editors on this suggestion, as I can't make that format correctly in Latex.

106
timesteps --> time-step lengths?

Changed.

132 (& hereafter)
[typographical]
1e-4 to $10^{-4}$ (imagine it being typeset) and so on

Changed.

133
Because you are discussing the workings of LEMs, might it be useful to consider the findings of Kwang & Parker (2017), that fluvial-only LEM outputs are scale invariant with m/n=0.5? I am wondering if this might be generally helpful when discussing the applicability of your results: despite your careful consideration of standardized initial conditions, you have also picked a parameter set that should be scale invariant.

I'm unsure what is being asked here.

136 (Eq. 3)
Could you pull the "v" outside of the differential operator?
Alternatively, you could change the "d"s to "\Delta"s, which would make this become a fraction instead of an operator.

Changed.

150-160
[self-narration; no response needed]
I am anxious about how the stability of your time steps might relate to your findings here, especially because only a subset of the models tested will be used for certain time steps. Because of how drainage-area changes can affect

knickpoint celerity, and drainage-capture events can occur instantaneously, I am likewise wondering if this could turn into the major time-step dependence. I'm continuing to read with this in mind.

152-153
Inline citation; consider "shown by"
Changed.

183 (& following)
Why "100,000"? And this is presumably years. It feels arbitrary, so a small note could help.

Explained below the equations.

190
Could this more precisely be called "maximum temporal change in elevation" (as opposed to "temporal change in maximum elevation")? Then the English would follow the math closely.

Changed.

195
A pair of nitpicks, one more real and one more convention.
The more "real" one is that dissolved, colloidal, etc. load are not considered to be "sediment". Therefore, perhaps you could consider noting the temporal change in total material removed from the landscape. This says what you are going for while removing the implicit assumption that eroded material becomes sediment.
Second, "flux" is mathematically and traditionally used as the transfer of a quantity across a plane, such that its units are [thing]/[area*time]. This is not how the sedimentologits & al. use it, or the climate scientists. So that's why this is a nitpick.

We changed this a bit, although we are using what we think is standard language in landscape evolution models, and we aren't sure what else we would call it.

214-218 (please ignore; including for narrative of my thinking during the review process)
This threshold seems arbitrary. Might it be better to consider a time scale that emerges naturally from the model outputs, e.g., finding an exponential decay time scale and defining some multiple of this?
On second thought, I scrolled down to your plots. I suppose they aren't all that exponential!

224 (Eq. 10)
Could you define "L"? (I've double-checked and hope I'm not just missing it!) I'm guessing that it is some kind of length scale -- perhaps half of the domain width?

Done.

Table 1 + Figure 1: This is a small thing, but I wonder if you could place the experiments within both of these in the same order. It would help in interpreting the results.

Changed.

Figure 3: Some dangling text has been left in the caption at the end.

Fixed

334-335 (also a bit of my own thinking, use/don't use at will)
I agree about this being the minimum response time, as (with your above note) the channels might not until later extend all the way up to the drainage divide with a steady-state basin area. I think that one other implication of this important point (though perhaps also beyond the scope of your article here) is the oft-invoked but seemingly mythical beast of channel headward propagation, used often in arguments of drainage integration. Knickpoints can propagate quickly upstream, but rearranging the channel head where drainage area --> 0 can be quite slow.

**Reviewer 3**

General comments
Dr. Gasparini and co-authors presented, as revealed by the title, issues with calculating time to steady state in landscape evolution models based on various time steps and landform metrics and compared that time between different LEMs. Compared to the original version, this revised manuscript is more centered on one specific theme (i.e. time to steady state) for model intercomparisons. I appreciate how the sections are now structured, with sufficient (but not too much) details of the theory, LEMs, and their methodology presented. Their results are also clearly explained and overall convincing. I especially agree with the implication of their work that "… it is critical to both report the metric being used and the threshold value for that metric to assess steady state (line 302)". I believe this work will encourage further investigations on other issues related with benchmarking LEMs, and this work provides a nice reference for those future studies. That will surely benefit the whole geomorphologic community. With all these thoughts, I recommend publishing this work after very minor revision.

We kept the line that this reviewer particularly liked.

Specific comments
You used different landform metrics and time steps and a threshold to calculate the time to steady state. I am curious about the values of those metrics (max Z, mean Z, local max Z, flux) at steady state and how different they are between models. Could you add those values in the subplots of Figure 1?

We did not add these values, simply because we were trying to stay focused and create a "short" communication.

Technical corrections
Figure 1: change the unit of dt to be yrs, making it more consistent with the main text. Same applies to Figure 3.

Because the values are so long we left them as My. This makes the plots easier to read, at least to us.

Line 315: cite Li et al 2018 in bracket.

That sentence was deleted.

---

## Author Response (AR3)

29 Aug 2024

Dear Wolfgang,

Thank you for your patience and persistence with this one. I'm guessing you are as happy to have this over as we are. It didn't go as I'd hoped, but maybe someday I will find the energy to try again.

We have updated all our figures to (hopefully) make them more readable. We also added a new first figure that illustrates model topography at three times and the network change between these times. The only changes to the text were the figure captions, text to incorporate the new figure, and a very few small changes to sentences to make them more readable. (I couldn't help myself!)

Thank you for your volunteer work as an AE. It is so valuable to our community.

Sincerely,
Nicole Gasparini
Tulane University